

# Assessing spatial heterogeneity of active layer thickness over Arctic-foothills tundra through intensive field sampling and multi-source remote sensing

Jinyang Du[1], K. Arthur Endsley[1], Kazem Bakian Dogaheh[2, 3], John S. Kimball[1], Mahta Moghaddam[3], Thomas A. Douglas[4], Asem Melebari[3], Sepehr Eskandari[3], Jinhyuk Elisha Kim[5], Jane Whitcomb[3], Yuhuan Zhao[3], Sophia Henze[3]

[1]Numerical Terradynamic Simulation Group, W.A. Franke College of Forestry & Conservation, University of Montana, Missoula, MT, 59812, USA
[2]Department of Earth and Environmental Sciences, University of Michigan, Ann Arbor, MI 481019, USA
[3]Ming Hsieh Department of Electrical Engineering, University of Southern California, Los Angeles, CA 90089, USA
[4]U.S. Army Cold Regions Research and Engineering Laboratory, Fort Wainwright, AK, 99703, USA
[5]Department of Earth System Science, University of California, Irvine, California, 92697, USA

*Correspondence to*: Jinyang Du (jinyang.du@ntsg.umt.edu)

**Abstract.** Active layer thickness (ALT) is an essential climate variable for monitoring permafrost degradation. The ALT deepening can lead to increased greenhouse gas emissions, altered hydrology and ecology, infrastructure damage, and a positive climate feedback. Quantifying ALT spatial heterogeneity remains challenging due to the influence of localized variations in terrain, microclimate, snow/soil properties, vegetation cover, and surface disturbances. It is also unclear how local ALT patterns (e.g., sub-meter to 10 m) and mechanisms scale up to broader landscape footprints (e.g., 10 -1000 m) represented from global satellite observations and Earth system models. We assessed ALT spatial heterogeneity in the Arctic-foothills tundra within the Northern Slope of Alaska through intensive field sampling over four 90 m x 90 m plots, combined with multi-source remote sensing and machine learning (ML). Analysis using field observations and ML revealed that vegetation, surface wetness, subsurface rocks, and micro-topography exert strong influence on 5-m ALT variations, whereas terrain controls dominate (~65% contribution) at coarser 10-m spatial resolution. By leveraging cm-level optical-infrared drone imagery, we further generated 0.1-m ALT maps over a larger 5 km x 5 km region and examined ALT scaling effects. Our analysis showed a quadratic relationship in scale-dependent uncertainties, characterized by a rapid increase in uncertainties at the sub-meter level (e.g., RMSE normalized by the standard deviation of 0.1m ALT climbed by ~10%), followed by another 10% increase from 1 m to 30 m resolution, and more conservative error increase (~5%) from 30 m to 1,000 m scale. Our study allows for improved interpretation of remote sensing and process-based ALT simulations for the changing Arctic by clarifying scale-dependent uncertainties and underlying mechanisms.

## 1 Introduction

Permafrost in the northern high latitudes is undergoing rapid changes driven by enhanced regional warming at roughly four times the mean global rate (Rantanen et al., 2022). Complex environmental changes that accompany degrading permafrost include widespread earlier spring thawing and lengthening of the thaw season, shifts in seasonal snow cover properties, deepening active layers (layer on top of permafrost that undergoes seasonal freeze/thaw) and contrasting wetting and drying patterns, vegetation greening and browning, ground surface deformation, and increasing disturbances (Du et al., 2019; Heijmans et al., 2022; Foster et al., 2022). In addition, thawing permafrost could potentially mobilize a vast reservoir of soil



organic carbon previously stabilized in perennially frozen ground, resulting in accelerated soil decomposition and greenhouse
gas emissions that may further amplify global warming (Biskaborn et al., 2019; Turetsky et al., 2020; Schuur et al., 2022).

Active layer thickness (ALT), defined as the maximum summer thaw depth above near-surface permafrost (Zhang et al., 2005, Schaefer et al., 2015), is an essential climate variable for monitoring permafrost degradation (Michaelides et al., 2019). Accurate mapping of ALT spatial distributions and temporal dynamics is critical for understanding impacts of climate change
and disturbance on the permafrost energy balance, terrestrial water budget, organic matter decomposition, and land-atmosphere carbon exchange (Zhang et al., 2005; Schaefer et al., 2015; Liu et al., 2024). ALT spatial patterns are closely linked to ambient air temperature at regional scales while demonstrating large spatial heterogeneity at local scales due to complex interactions with precipitation, vegetation, slope, aspect, soil and snow properties, topography, and disturbance (Leibman et al., 2012; Widhalm et al., 2017; Loranty et al., 2018; Chen et al., 2019).

Despite the importance of permafrost landscapes in monitoring and projecting Arctic-boreal system changes, there is only limited understanding of local-scale (e.g., sub-meter to 10 m) ALT patterns and underlying driving factors over the highly heterogeneous permafrost regions due to a paucity of high-resolution ALT measurements. It is also unclear how local ALT patterns and its associated governing mechanisms scale up to the broader landscape when quantified at coarser spatial
resolutions (e.g., 10 m-1000 m) using global satellite observations and regional process models (Obu et al., 2019; Yi et al., 2020). However, better understanding of the local-scale patterns, mechanisms, and their scaling effects are needed to clarify permafrost vulnerability and associated feedbacks to climate change, and to improve representation of the hydrological, ecological, and biogeochemical processes of Arctic ecosystems in global Earth system models as well as remote sensing inversion algorithms (Koven et al., 2013; Chen et al., 2019; Hantson et al., 2025).

Remote sensing of ALT at local scales ranges from relatively direct electromagnetic retrievals from low frequency ground penetrating radar (GPR) to more indirect measures of soil freeze-thaw (FT) driven land surface deformations from Light Detection and Ranging (LiDAR) and Interferometric synthetic aperture radar (InSAR) measures (Kneisel et al., 2008; Schaefer et al., 2015; Du et al., 2019). Low-frequency (e.g., L- and P-band) microwave measurements are capable of penetrating through
vegetation and soil layers and show strong potential for mapping active layer properties, including soil moisture and FT dynamics, organic matter content, and ALT (Tabatabaeenejad et al., 2014; Bakian-Dogaheh et al., 2025). For example, retrieval algorithms were developed by exploiting airborne P-band radar data for ALT mapping in Alaska (Chen et al., 2019), which showed favorable retrieval accuracy for ALT up to ~60 cm. Another study on post-fire recovery in tundra regions further confirmed the unique sensitivity of low-frequency airborne radar in detecting the relatively fine-scale heterogeneity in active
layer conditions and post fire recovery that may be missing from coarser-scale and higher-frequency satellite observations and model simulations (Yi et al., 2021). More recently advanced physics-based computational radar retrieval algorithms have been



developed to map the active layer soil organic matter and soil moisture profile of the Arctic-foothills tundra using P-band radar data (Bakian-Dogaheh et al., 2022).

Besides direct microwave sensing of soil profiles, ALT can also be indirectly inferred from optical vegetation observations and relatively high-frequency radar backscatter signals considering the ALT dependence on surface conditions (Kelley et al., 2004; Gangodagamage et al., 2014; Widhalm et al., 2017). Recent machine-learning based studies using airborne hyperspectral imaging data achieved meter-level ALT predictions over Interior Alaska with favorable accuracy (Zhang et al., 2021). Thus, complementary information available from multi-sensor and multi-scale remote sensing provides the basis for developing

more effective models and ALT retrievals capable of resolving local patterns and environmental linkages spanning Arctic-boreal permafrost landscapes (Brodylo et al., 2024; Hantson et al., 2025).

The objectives of this study were to: 1) assess ALT spatial heterogeneity and scaling properties from 0.1-m to 1000-m resolution within the tundra foothills region of the North Slope of Alaska; and to 2) clarify the underlying environmental

controls on ALT patterns manifesting at different scales and the potential influence of spatial resolution on regional permafrost remote sensing and modeling uncertainty.

## 2 Study Region

Warming is promoting widespread permafrost thaw across Alaska, where annual temperatures are projected to rise by more than 5.6 ºC by the end of the century (Sun et al., 2015). Our study focused on the Imnavait Creek (68.6167° N, 149.3167° W)

area within the Alaskan North Slope tundra foothills region where deepening ALT was observed from both in-situ measurements at the Circumpolar Active Layer Monitoring (CALM) sites (https://www2.gwu.edu/~calm/data/north.htm) and regional ALT records (Liu et al., 2024). The study area is comprised of gently rolling topography underlain by continuous permafrost with ALT ranging from ~25 to 100 cm in depth (Schramm et al., 2007). The local study area consisted of four intensively sampled field plots (Plot 3, Plot 4, Plot 5, and Plot 6 in Fig. 1; 90 m × 90 m each) distributed across an elevation

gradient along west facing hill slopes, and surrounded by a larger (5 km by 5 km) landscape domain used for analyzing ALT scaling effects (Fig.1). The field plots are dominated by tussock-forming sedges and moss-lichen mats, along with scattered dwarf shrubs, and riparian grasses (Fig. 1-2). Plot 3 and Plot 4 are located on west-facing downhill slopes characterized by gradual elevation changes, small water tracks, and scattered glacier erratics. Plot 5 is located at a valley bottom and is partitioned by Imnavait Creek, while Plot 6 is characterized by widespread subsurface rocks (Fig. 1). All maps in the study

were plotted in the Canada Albers Equal Area Conic projection.





**Figure 1: The study region encompasses an Arctic tundra area (68.6167°, -149.3167°; red dot in the inset) in the northern foothills of the Brooks Range, Alaska. The region consists of four intensively sampled plots (Plot 3, Plot 4, Plot 5, and Plot 6; 90 m x 90 m each) with their corresponding true-color RGB (red-green-blue) drone images displayed alongside, and surrounded by a larger 5 km by 5 km study region (red rectangle) used for analyzing ALT scaling effects.**



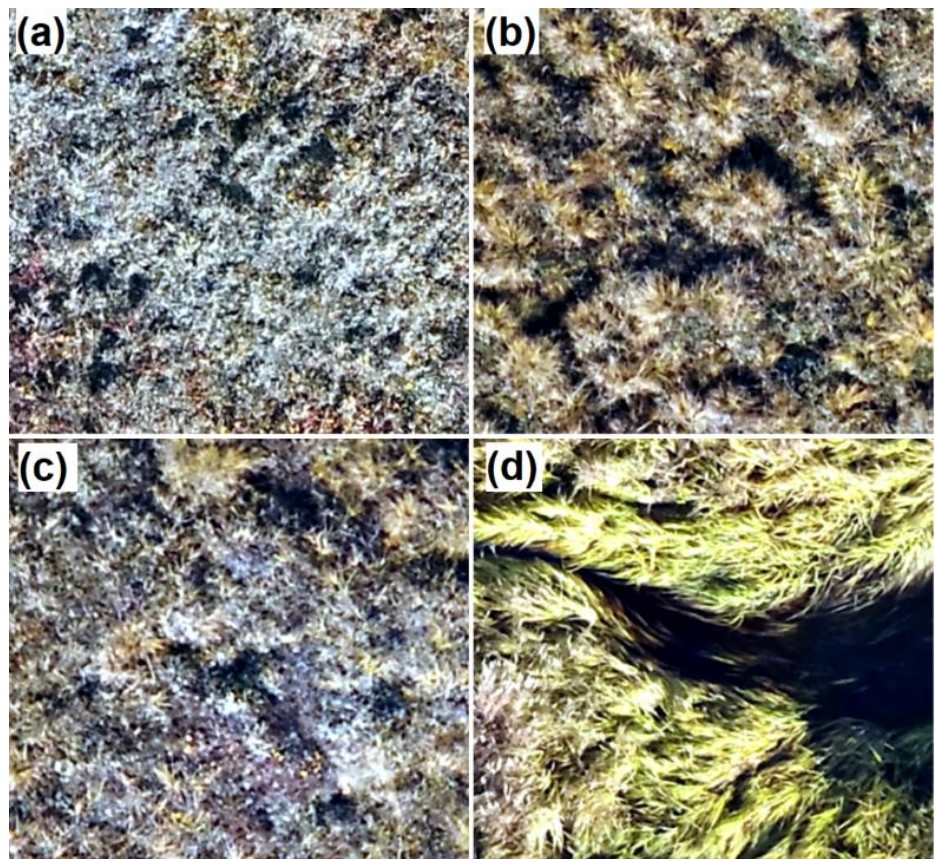

**Figure 2: The study region contains characteristic tundra vegetation types, including dwarf shrubs (a), tussock-forming sedges (b), moss-lichen mats (c), and riparian grasses along watercourses (d). The images (0.7 cm resolution) were acquired from a drone-based RGB camera.**

The surrounding study region (Fig. 1) has rolling terrain, with elevations ranging from ~750 m to 980 m. Vegetation and soil patterns vary with slope, aspect, and drainage conditions, exhibiting a patchy distribution across the study domain (Hinkel and Nelson, 2003). The region is also characterized by diverse glacial landforms, including deposits, stream networks, and bedrock outcrops (Hinkel and Nelson, 2003).

## 3 Data and Methods

### 3.1 Data Sets

Multi-scale ALT mapping was performed using probe-based field measurements, high-resolution (centimeter level) drone-based optical-Infrared (IR) acquisitions, concurrent airborne L-band radar observations, Sentinel-2 satellite images, and ancillary digital elevation model (DEM) data.





### 3.1.1 Field Sampling

Ground sampling activities were conducted within the field plots from August 16-23, 2024, which included probe-based ALT measurements, soil coring, and surface roughness measurements. Each plot (Fig. 1) contains 289 grid cells (containing 17 x 17 subpixels with 5 m x 5 m each) excluding a 2.5 m buffer zone along the plot edges. Probe-based ALT sampling was performed over every (or every other) grid cell. Three measurements (by inserting the probe in the active layer vertically until refusal) were taken randomly within each grid cell, unless the initial two measurements were less than 1.27 cm apart. For Plot 5, samples were collected for every grid cell not covered by open water, whereas for Plot 6, sampling only covered about half of the area due to widespread shallow rocks. The readings were averaged to represent the ALT conditions for a given 5-m grid cell. Overall, approximately 1700 measurements were made representing ca. 600 grid cells across the plots. Additional notes describing surface features were included for grid cells dominated by shrubs, standing water, running water, and subsurface rocks. Field measurement protocols followed previously established methods within the NASA Arctic Boreal Vulnerability (ABoVE) field campaign (Schaefer et al., 2021; Bakian-Dogaheh et al., 2022).

### 3.1.2 Remote Sensing Data Sets

For understanding ALT spatial heterogeneity, multi-scale remote sensing measurements were collected from drone, piloted aircraft, and satellite platforms. Aerial surveys included drone-based optical-IR observations, and airborne L-band radar observations from the NASA Uninhabited Aerial Vehicle Synthetic Aperture Radar (UAVSAR). The field and airborne data were collected under the ABoVE campaign, an intensive decadal field campaign focused on improving our understanding of climate-related impacts on Arctic and boreal systems in western North America (Miller et al., 2019).

For this study, high-resolution ALT mapping relied on drone-based optical-IR observations from a RGB (red, green, blue) camera (0.7 cm resolution), a Micasense multispectral (blue, green, red, red-edge, near IR bands) camera (2.7 cm resolution), and a DEM (1.2 cm resolution) derived from the RGB imagery. These images were captured over each plot on September 4th, 2024 by the ToolikGIS team from the Toolik Field Station of the University of Alaska Fairbanks. For capturing vegetation conditions and surface water signals, the Normalized Difference Vegetation Index (NDVI; Tucker, 1979) and the Normalized Difference Water Index (NDWI; McFeeters 1996) were calculated from the drone multispectral imagery. In addition, aspect and slope were calculated from the DEM. The drone-based optical images and products were aggregated to both 0.1 m and 5 m resolutions for supporting the machine learning (ML) and scaling analysis (section 3.2).

For ALT mapping over the surrounding region at coarser spatial resolution, ESA Sentinel-2 satellite optical-NIR observations under clear-sky (cloud cover < 20%) conditions were composited over the summer months (June 15 – August 15, 2024) in Google Earth Engine (GEE). The NDVI and NDWI were derived using the Sentinel-2 observations at 10-m resolution. For



understanding terrain controls on the regional ALT distribution, elevation, slope, and aspect data were derived from the ArcticDEM (Porter et al., 2018) and aggregated to 10 m resolution.

To examine the possible contributions of airborne radar observations to ALT retrievals, the ML model for 5-m ALT mapping was augmented with additional UAVSAR L-band VV, HV and HH backscatter observations in a separate test. The UAVSAR data were acquired over the Imnavait study region on August 21, 2024 concurrent with the field measurements (Miller et al., 2024). However, the airborne radar data were not used in the scaling analysis, which primarily relied on drone optical imagery for providing cm-level details and Sentinel-2 observations for enabling large-area analysis.

**3.2 Constructing Multi-scale ALT Maps**

A multiscale approach was used for characterizing ALT spatial heterogeneity, understanding environmental driving factors, and quantifying scale-dependent uncertainties in remote sensing retrievals. Data-driven Random Forest (RF) models were trained for re-constructing ALT patterns over the intensely sampled plots (5-m resolution) and applied using multi-source remote sensing inputs for generating the respective ALT maps at 0.1 m for the plots and 10 m over the surrounding study
region (Figure 3).

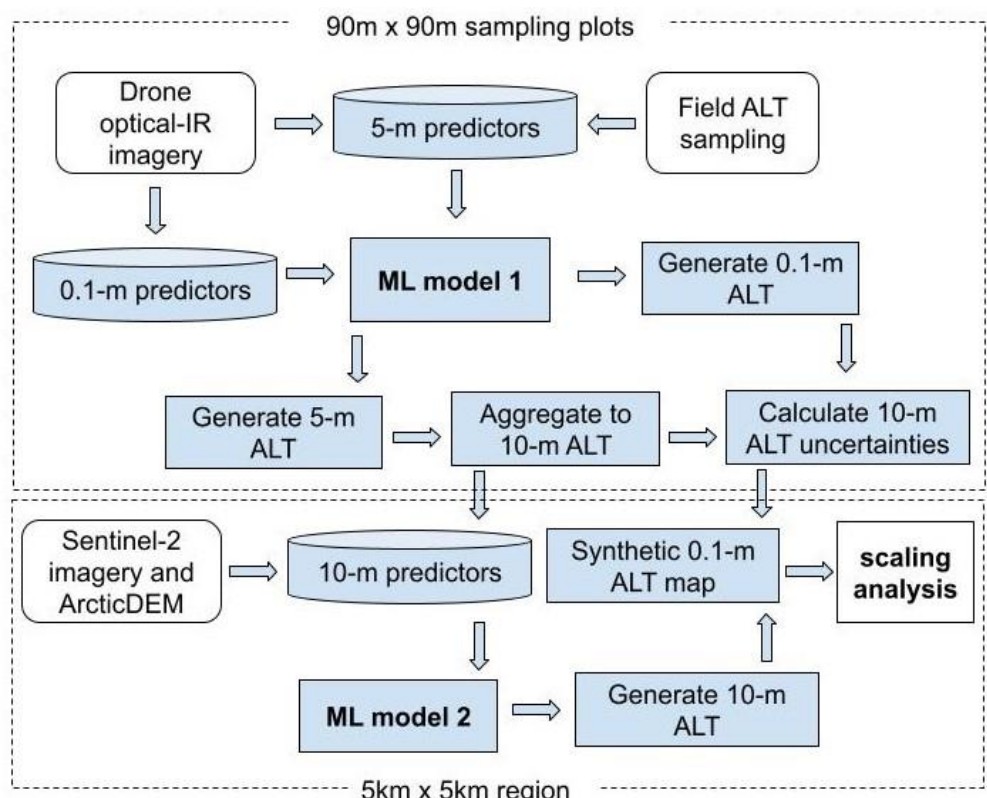

**Figure 3: An overall flowchart of the machine learning (ML) based active layer thickness (ALT) mapping and analysis.**





The RF method has been widely used in enhanced remote sensing of land surface parameters such as soil moisture, vegetation
optical depth, and ALT (Zhang et al., 2021; Du et al., 2024). The RF tree ensemble yields final regression results by combining
individual predictions, resulting in improved accuracy and stability compared to a single tree (Breiman, 2001). Compared with
alternative ML methods, the RF demonstrates resilience to data noise and overfitting (Breiman, 2001).

For generating the baseline 5-m ALT maps over the sampling plots, the first RF model was trained using the local high-
resolution drone optical-NIR imagery, which was aggregated to a coarser (5 m) resolution consistent with the spatial
representativeness of the field sampling. The RF target variable is ground-sampled ALT, and the model predictors (Table 1)
were selected from different combinations of observations and indices. The model was trained using 80% of the data and
validated using 20% of the data. Different sets of hyper-parameters (number of trees 10, 25, 50, 100 and minimum leaf
population 1, 3, 5) were employed for tuning and evaluating the regression trees using GEE. The root mean square error
(RMSE) and correlation coefficient (R) were calculated for performance evaluation. To reduce overfitting risks, the smaller
(larger) value of tree number (minimum leaf population) was selected for prioritizing a relatively simple tree/forest structure
if two hyper-parameter values led to similar performance (e.g., R difference less than 0.005). The optimized RF hyper-
parameters are 25 trees and a minimum leaf population of 3.

The 5-m ALT outputs for the sampling plots from the first RF model were aggregated to 10-m resolution and used to train the
second RF model. The second model was developed for regional ALT mapping by leveraging operational satellite data as
model predictors. The predictors (Table 1) were selected from terrain features and Sentinel-2 observations. Following a similar
dataset splitting and hyper-parameter tuning process, the best performing model was built with a relatively simple structure
(10 trees, 1 minimum leaf population).





**Table 1. Machine learning predictors for 0.1-m, 5-m, and 10-m ALT mapping and scaling analysis.**

| Predictors for 5-m and 0.1-m ALT mapping over sampling plots | | | Predictors for 10-m ALT mapping over 5 x 5 km region | | |
|---|---|---|---|---|---|
| Name | Spatial resolution | Drone sensor | Name | Spatial resolution | Data source |
| R | 0.7 cm | RGB Camera | Elevation | 2 m | ArcticDEM |
| G | 0.7 cm | RGB Camera | Slope | 2 m | ArcticDEM |
| B | 0.7 cm | RGB Camera | aspect | 2 m | ArcticDEM |
| Red edge | 2.7 cm | Multispectral Camera | NDVI | 10 m | Sentinel-2 |
| Near infrared | 2.7 cm | Multispectral Camera | NDWI | 10 m | Sentinel-2 |
| NDVI | 2.7 cm | Multispectral Camera | | | |
| NDWI | 2.7 cm | Multispectral Camera | | | |
| Aspect | 1.2 cm | RGB Camera | | | |
| Slope | 1.2 cm | RGB Camera | | | |
| Note: Drone images were acquired on 09/04/2024, and the predictors were processed to 0.1-m and 5-m resolutions for the respective ALT mapping. | | | Note: Terrain data were averaged to 10-m resolution; Satellite images were aggregated over the 2024 summer period from 06/15 to 08/15. | | |

### 3.3 Assessing ALT spatial heterogeneity

For studying the ALT spatial heterogeneity observed at different scales, the first RF model was applied to the drone-based optical-NIR observations aggregated to 0.1 m resolution. The 0.1-m ALT maps were generated as a fine-scale benchmark for evaluating uncertainties in the coarser spatial resolution ALT maps. The 0.1-m ALT maps for the sampling plots were aggregated to 10-m resolution. By comparing the differences between the two maps for each 0.1-m pixel, the estimated RMSE of the 10-m product (referred hereafter as $RMSE_{10m}$) was obtained.


For understanding the ALT variations over larger regions beyond the limited sampling plots, the second RF model was applied using Sentinel-2 and DEM data over the surrounding 5 km by 5 km area for generating the 10-m ALT regional map. A synthetic 0.1-m ALT map over the surrounding area was then generated using the 10-m ALT map derived from the second RF model and perturbed with noise with a normal distribution and $RMSE_{10m}$ standard deviation.


Multiple ALT regional maps with spatial resolutions ranging from 0.2 to 1000 m were generated by aggregating the 0.1-m pixels of the synthetic map. The aggregation intervals were set as 0.1 m, 1 m, 10 m, and 100 m for deriving the respective





maps at sub-meter, meter, decameter, and hectometer resolutions. The uncertainties of a given coarser-resolution map relative to the 0.1-m benchmark were then obtained by calculating their differences over each 0.1-m pixel and measured using two metrics, namely, the relative RMSE normalized by the ALT standard deviation of the 0.1-m data (RMSE/Stdev) and the relative RMSE normalized by the mean ALT (RMSE/mean).

## 4 Results

### 4.1 Plot and regional ALT mapping

#### 4.1.1 Field sampling and multi-source ALT mapping for the sampling plots

The ALT values sampled across all plots have a mean of 45.9 cm and a standard deviation of 11.9 cm. Relatively shallow ALT (39.3 cm) was typically found in areas dominated by non-riparian shrubs, while deeper ALT occurred in soils near standing water (61.0 cm), along creeks (78.2 cm), and around rocks (70.0 cm).

The first RF model was able to reproduce the sampled ALT at 5-m resolution with a RMSE of 6.53 cm and strong correlation (0.78). Among all the predictors, slope (17.40%), red-edge reflectance (14.05%), aspect (13.44%), and NIR reflectance (12.64%) were the most important features contributing to ALT predictions. The contributions from other predictors including green band (9.91%), blue band (9.43%), NDWI (8.48%), red band (7.41%), and NDVI (7.24%) were also important.

The RF model was then applied to the same set of predictor features, but at finer (0.1-m) spatial resolution. The resulting 5-m and 0.1-m ALT maps were compared with the field measurements (Fig. 4). The 5-m ALT maps (Fig.4b, 4e, 4h, 4k) captured the primary ALT patterns observed from the field measurements (Fig. 4a, 4d, 4g, 4j) including elevated ALT along the perimeter of Plot 3, consistently low ALT throughout Plot 4, high ALT values following the water tracks in Plot 5, and generally deep ALT in Plot 6. The 0.1-m results revealed significantly finer variations (Fig. 4c, 4f, 4i, and 4l), resolving ALT patterns associated with small water tracks (e.g., Fig. 4i), discrete vegetation patches (Fig. 4c, 4l), and widespread clusters of underlying rocks (Fig. 4l) that were not discernible at 5-m resolution.





**Figure 4: Comparisons of ALT spatial patterns derived from field measurements (a, d, g, j), 5-m machine learning outputs (b, e, h, k), and 0.1-m machine learning estimates (c, f, i, l) for the intensively sampled plots.**



### 4.1.2 Satellite-based ALT mapping over the surrounding region

For the second RF model, high accuracy (RMSE 1.63 cm, R 0.97) was achieved for the 10-m ALT predictions over the sampling plots. Terrain factors act as the most important control on the ALT distributions at 10-m resolution (elevation 29.88%, aspect 17.85%, slope 17.16%), while vegetation (NDVI 19.85%) and surface wetness (15.26%) conditions are relatively less important.

The model was then applied using Sentinel-2 observations over the larger region surrounding the sampling plots. The resulting regional ALT patterns are strongly influenced by local topography, which is consistent with the RF contribution analysis. The higher-elevation portions of the rolling terrain and gentle hill slopes generally have deeper ALT compared with the steeper slopes (Fig. 5a, 5d) whereas aspect has a secondary role (Fig. 5c).

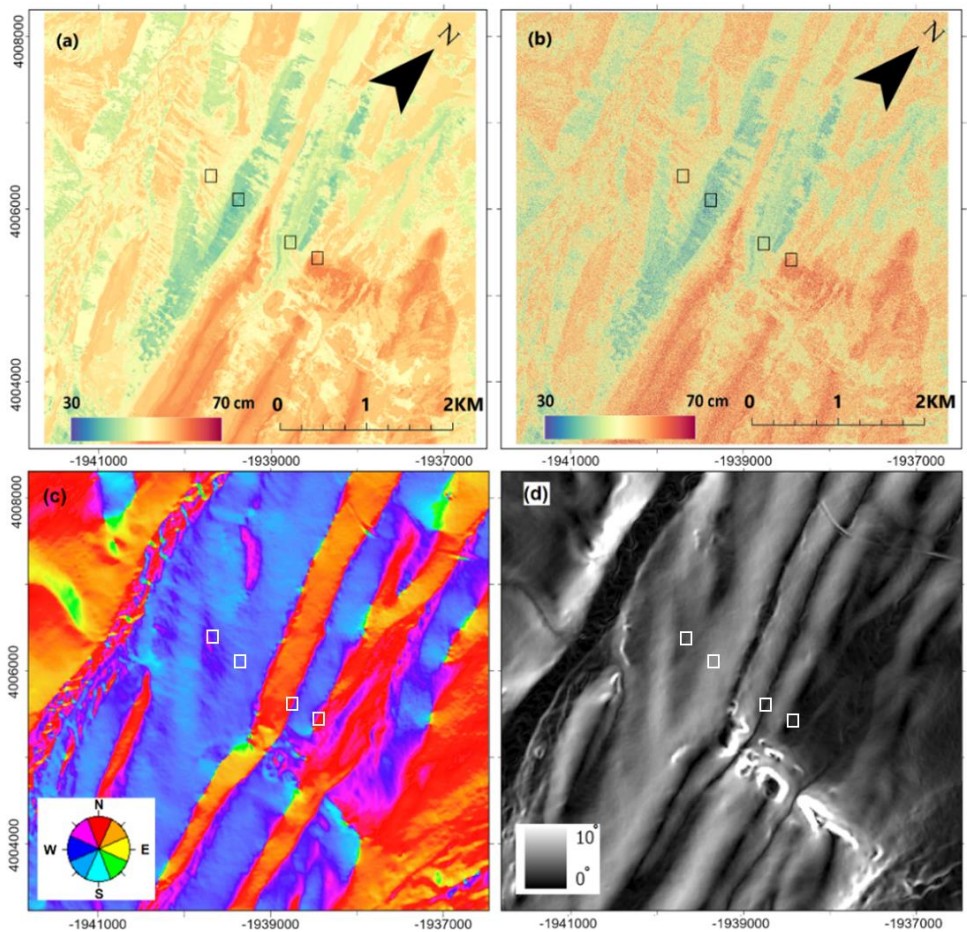

**Figure 5: Comparisons between ALT distributions at 10-m resolution derived from machine learning (a), the synthetic ALT map at 0.1-m resolution (b), terrain aspect (c), and slope (d) over a larger 5 km x 5 km area surrounding the sampling plots (black/white rectangles).**



## 4.2 Scaling Effects Analysis

The uncertainties associated with coarser-resolution results were calculated by comparing with the synthetic 0.1-m ALT map (Fig. 5b). Uncertainties increase at coarser spatial resolution, with the most rapid growth occurring at finer scales (e.g., sub-meter to 1 m), and slower changes occurring at coarser pixel sizes (e.g., 100 - 1000 m). The increase in ALT uncertainty at coarser spatial scales generally follows a quadratic functional form with best fit represented by a second-order polynomial; however, the accelerated error increase at sub-meter scales is better characterized by a different second-order polynomial (Fig. 6). The normalized uncertainties (RMSE/Stdev) range from ~68% to 95%, which suggests a majority of the 0.1-m variations are likely smoothed out when observed or modelled at coarser resolutions (e.g., 100-1000 m) (Fig. 6). However, the RMSE magnitudes are relatively small (~9.6% to 13.1%) as compared to the 0.1 m ALT mean (Fig. 6).

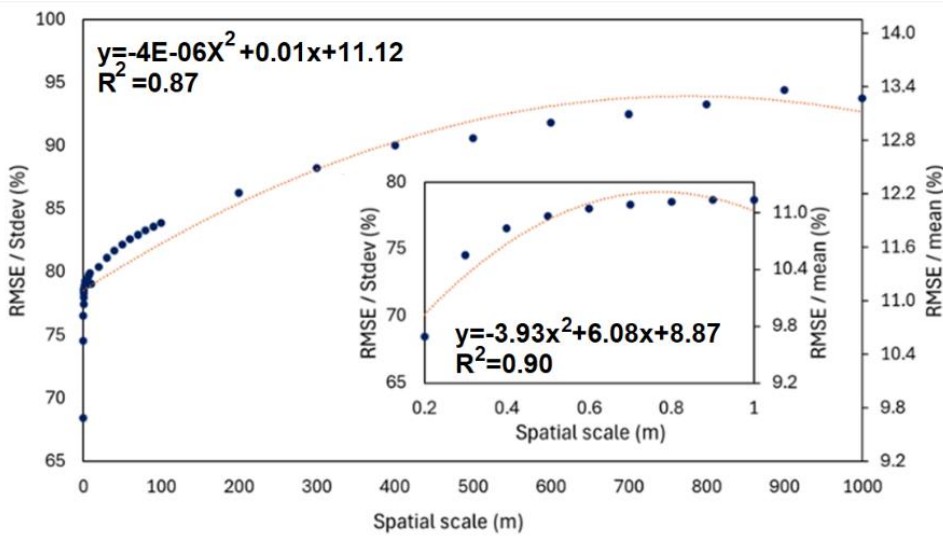

**Figure 6: ALT uncertainties change with spatial resolutions from 0.2 to 1000 m with a detailed subset analysis from 0.2 m to 1 m (inset). The uncertainties were determined using the 0.1-m ALT benchmark and normalized by the standard deviation (left vertical axis) and mean (right vertical axis; fitted polynomial function) of the 0.1-m ALT values.**

## 5. Discussion

## 5.1 Key Factors Controlling ALT Spatial Variations

While regional ALT dynamics at kilometer or coarser scales are primarily governed by air and surface temperatures (Gangodagamage et al., 2014; Peng et al., 2023), local ALT variations at finer scales (sub-meter to 100 m) are highly complex and influenced by a range of climatic and environmental factors (Gangodagamage et al., 2014). Here, vegetation conditions, snow cover properties, soil texture, soil moisture, surface water bodies, groundwater flow, micro-topography, and disturbances



collectively determine the local-scale ALT pattern (Gangodagamage et al., 2014; Grünberg et al., 2020; Rushlow et al., 2020; Clayton et al., 2021).

Consistent with previous studies, our field observations (section 4.1.1) confirmed that deeper ALT is most common in areas with standing water or adjacent to creeks, where wet conditions enhance soil thermal conductivity in foothills tundra (Grant et

al., 2017; Clayton et al., 2021). Relatively deeper ALT was also found in the vicinity of subsurface rocks (e.g., within 1-m distance), whose high thermal conductivities facilitate heat propagation and summer thawing (Bonnaventure et al., 2013). Relatively shallower ALT was recorded over the shrubs, which likely cool the ground in summer through canopy shading (Lawrence and Swenson, 2011) and in winter through thermal bridging effect (Domine et al., 2022). Similar to shrubs, a thick moss layer may also slow active layer thaw through its insulating capacity (Schuuring et al., 2024), though no specific

descriptions of the moss layer were made in our sampling.

For the ML- and drone-based analysis, the surface features including vegetation, water bodies, and soil properties determined from the multi-spectral reflectance and optical-NIR indices, all showed important contributions to the 5-m ALT predictions over the sampling plots (7.24% to 14.05%; section 4.1.1). In addition, micro-topography information including slope (17.40%)

and aspect (13.44%) are among the most important factors shaping the local-scale ALT variations, and consistent with a previous study focusing on coastal tundra (Gangodagamage et al., 2014). It is noted that additional radar-based observations from UAVSAR helped with enhancing the performance of the first RF model marginally (e.g., R from 0.78 to 0.81), but were not used in the subsequent scaling analysis (section 3.1.2).

For the ML- and satellite-based analysis, terrain factors (elevation, slope, and aspect) collectively dominate the ALT predictions (64.89% contribution) at 10-m resolution. The broad ALT patterns over the surrounding region (Fig. 5a) largely align with terrain-driven variability (section 4.1.2), as also observed in a previous study (Hinkel and Nelson, 2003).

## 5.2 ALT Scaling effects

Quantifying ALT uncertainties associated with varying spatial scales of observation and modeling is needed for determining

the validity of remote sensing based ALT products, improving permafrost models and simulations, and understanding future climate-ecosystem-hydrology feedbacks (Hantson et al., 2025). Our analysis revealed a rapid increase in ALT spatial uncertainty at the sub-meter scale (e.g., RMSE/stdev climbed by ~10%), followed by another 10% increase from 1 - 30 m resolution, and more conservative error growth (~5%) from 30 m to 1000 m resolution. Additional variogram analysis on the 0.1 m ALT results further showed the range or correlation length can be as small as 0.86 m (e.g., upper-left part of the sampling

area of Plot 3; Fig. 4c), suggesting that fine-structure active layer features can be quantified by the methodology. High resolution (e.g., meter or sub-meter level) observations are therefore needed for quantifying the small-scale ALT heterogeneity,



which is consistent with the findings from a discontinuous permafrost region in the Alaskan Seward Peninsula (Hantson et al., 2025).

On the other hand, the scaling effects follow a quadratic form represented by a second-order polynomial function (Fig. 6), which approaches greater error stability at coarser resolutions (e.g., 30 - 1000 m) and with relatively low RMSE/mean values (e.g., <13.4%). These results indicate that while fine-scale heterogeneities are smoothed out at coarser resolutions, broader ALT patterns remain detectable from relatively coarser satellite observations (e.g., from Sentienl-2, Hantson et al., 2025; MODIS, Liu et al., 2024) and with acceptable error margins to reveal critical active layer dynamics and inform regional process

models.

## 6 Conclusions

ALT is a sensitive indicator of the thawing Alaskan permafrost, which may be leading to increased greenhouse gas emissions, altered hydrology and ecology, infrastructure damage, and positive climate feedback. This study represents a multi-scale assessment of ALT spatial heterogeneity within the Arctic foothills of northern Alaska, a region characterized by continuous

permafrost cover and complex terrain. We employed a machine learning framework for ALT mapping ranging from local plot (0.1 m) to landscape (10 m) and regional (30 - 1000 m) levels using multiscale observations, including intensive field sampling, and drone, airborne and satellite remote sensing.

Our study showed that vegetation, soil, surface water, and topography are all key drivers of meter-scale ALT patterns, while

terrain topography becomes increasingly dominant in controlling ALT distribution at coarser scales. Scaling effect analysis further revealed a quadratic functional relationship describing the growth of scale-dependent uncertainties in ALT mapping, indicating rapid increase in uncertainty at sub-meter scales, moderate error increase at intermediate (1 - 10 m) scales, and more conservative growth at coarser (30 - 1000 m) scales representative of many global satellite and model based assessments. For areas where soil, vegetation, and terrain conditions are markedly different from the Arctic foothills tundra, their ALT scaling

effects can be independently quantified through similar approaches from this study.

*Code and Data availability.* The data that support the findings of this study are in the process of archival and will be openly available through the Oak Ridge National Laboratory Distributed Active Archive Center (ORNL DAAC). Code used in this study is available upon request.


*Author Contributions.* JD wrote the manuscript. JD, KAE,KBD,AM,SE,JEK collected field data for analysis. JD, KAE,KBD, JSK, MM, TAD contributed to the design and conceptualization. All coauthors contributed to writing and editing of the manuscript. All coauthors have read and agreed to the published version of this manuscript.





*Competing interests.* We declare no competing interests are present.

*Acknowledgements.* This work was conducted at the University of Montana and University of Southern California with funding from the National Aeronautics and Space Administration (80NSSC22K1238). The optical-IR drone images were collected by the Toolik GIS team. The UAVSAR images were provided through the NASA ABoVE airborne campaign.

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
