# Peer review of "Assessing spatial heterogeneity of active layer thickness over Arctic-foothills tundra through intensive field sampling and multi-source remote sensing"

_EGUsphere, 2025_

## Author Comment (AC1)

RC1: 'Comment on egusphere-2025-3236', Anonymous Referee #1, 25 Aug 2025

**Comment 1.0**

Du et al. have submitted an interesting paper that investigates the spatial heterogeneity of the active layer thickness for a region on the North Slope of Alaska. The study combines field sampling, remotely sensed imagery and modelling. The analysis considers the relative influence of various climate and environmental factors on active layer conditions at different spatial resolutions. The study has potential to contribute to improved prediction of active layer thickness in a warming climate. However, I do have some concerns and comments that should be addressed for the manuscript to be acceptable for publication.

**Response 1.0**

Thanks for the careful review and insights into the study! We revised our manuscript accordingly. Please check our point-by-point responses below. All revisions were marked in blue.

**Comment 1.1**

Several previous studies have considered the relative influence of environmental factors on the ground thermal regime through analysis of field data collected across environmental gradients and some of the conclusions in the manuscript are not new. Recent papers, including those by some of the coauthors of the submitted manuscript have also considered quantification of active layer at multiple scales in Alaska (e.g. Brodylo et al. 2024). A better description of the novelty and advancement in knowledge of the submitted manuscript compared to earlier studies would be beneficial.

**Response 1.1**

As the reviewer pointed out, recent studies have been focused on "quantification of active layer at multiple scales in Alaska" using multi-sensor remote sensing such as the recent study for Interior Alaska (Brodylo et al. 2024) and the one for Seward Peninsula (Hantson et al. 2025). Despite a similar focus on resolving ALT multi-scale patterns and their environmental linkages, our study (a) highlighted the ALT local patterns over the unique Arctic-foothills tundra environment using intensive field sampling and remote sensing, and (b) quantified the resolution-dependent uncertainties in ALT retrievals inferred from multi-source remote sensing. The study thus provided additional support for improved interpretation of the multi-resolution ALT products derived using remote sensing and process-based simulations for the changing Arctic. In the revision, we added the following in the Conclusions section.

"In sum, our study mapped the ALT local patterns over the unique Arctic-foothills tundra environment, and quantified the resolution-dependent uncertainties in ALT retrievals inferred from multi-source remote sensing. The study provided additional support for improved

interpretation of the multi-resolution ALT products derived using remote sensing and process-based simulations for the changing Arctic".

**Comment 1.2**

The manuscript would benefit from a better description of the study plots and the broader region considered, such as information on surficial materials, vegetation and topography. Since the focus of the paper is spatial heterogeneity, it would be useful for the reader to have information on the spatial heterogeneity of the factors that are considered in the analysis within the study plots and in the broader area considered.

**Response 1.2**

As suggested, we revised the Study Region section to include a general description of the surrounding region, and more detailed information for the study plots. The revisions are also presented below.

"Our study focused on the Imnavait Creek (68.6167° N, 149.3167° W) area within the Alaskan North Slope tundra foothills region (Fig. 1). The study region experiences a cold climate with a mean annual temperature of -7.4°C, ranging from an average of -17°C in January to 9.4°C in July (Schramm et al., 2007). Annual precipitation averages 340 mm, two-thirds of which falls as light rainfall during the summer (Schramm et al., 2007). The dominant vegetation consists of water-tolerant plants like tussock sedges and mosses, grasses and low shrubs (Schramm et al., 2007). Vegetation and soil patterns vary with slope, aspect, and drainage conditions, exhibiting a patchy distribution across the study domain (Hinkel and Nelson, 2003). The region is also characterized by diverse glacial landforms, including deposits, stream networks, and bedrock outcrops (Hinkel and Nelson, 2003). The region has gently rolling terrain, with elevations ranging from ~750 m to 980 m. Vegetation and soil patterns vary with slope, aspect, and drainage conditions, exhibiting a patchy distribution across the study domain (Hinkel and Nelson, 2003). The region is also characterized by diverse glacial landforms, including deposits, stream networks, and bedrock outcrops (Hinkel and Nelson, 2003). Increasing ALT was observed from both in-situ measurements at the Circumpolar Active Layer Monitoring (CALM) sites (https://www2.gwu.edu/~calm/data/north.htm) and regional ALT records (Liu et al., 2024).

The local study area consisted of four intensively sampled field plots (Plot 3, Plot 4, Plot 5, and Plot 6 in Fig. 1; 90m ×90m each) distributed across an elevation gradient along west facing hill slopes, and surrounded by a larger (5 km by 5 km) landscape domain used for analyzing ALT across different resolutions (Fig.1; Table 1). The field plots are dominated by tussock-forming sedges and moss-lichen mats, along with scattered dwarf birch shrubs, and riparian grasses (Fig. 1-2). An informal survey of the common species identified dwarf birch (*Betula nana*), alpine blueberry (*Vaccinium uliginosum*), black bearberry (*Arctous* spp.), crowberry (*Empetrum nigrum*), and *Arctostaphylos uva-ursi*, among other less-recognizable species. Our soil coring samples indicated high organic matter content of the topsoil (0-10 cm depth) for Plots 3, 4 and 5,

with the respective values of 77.3%, 75.1%, and 71.8%, contrasting with a relatively low value of 45.5% for Plot 6.

Plot 3 and Plot 4 are located on west-facing downhill slopes characterized by gradual elevation changes (10-15 m across the plot), small water tracks, and scattered glacial erratics. Plot 3 is crossed by multiple drainage features that are transverse to the Kuparuk River (Fig. 1). These features are evident in the topography as well as the vegetation, with alternating bands of dwarf willow, grasses, and tussocks. Standing water is present in many places, as are glacial erratics, particularly along the drainage channels. Plot 4 is more homogeneous in vegetation character and height, consisting mostly of grass tussocks and moss, with some larger glacial erratics present.

Plots 5 and 6 are on the east side of the access road and are more level than Plots 3 and 4. Plot 5 is located at a valley bottom and is partitioned by Imnavait Creek, which winds directly through the middle of the plot. Shallow drainage channels transverse to Imnavait Creek are found on the west side of the plot, while the east side is upland and drier. Vegetation cover in Plot 5 follows the typical mix of grasses, mosses, and sedges, with some moss forming humps about 20 cm tall, while shrub density and height increase close to the creek. Along Imnavait Creek there are taller (0.5-1.0 m) birch shrubs and more grasses. Plot 6 is characterized by widespread subsurface rocks (Fig. 1) and overall short-statured grass-sedge tussocks and mosses, lichens, small bunchgrasses, *Arctous spp.*, and alpine blueberry, interrupted by areas of bare rock and gravel. Bare areas are common in the center of the plot."

Table 1. Summary of surface and soil conditions for the sampling plots

| Name                   | Plot 3                              | Plot 4                                               | Plot 5                                                             | Plot 6                         |
|------------------------|-------------------------------------|------------------------------------------------------|--------------------------------------------------------------------|--------------------------------|
| Elevation              | 810 m                               | 846 m                                                | 855 m                                                              | 875 m                          |
| vegetation             | dwarf willow, grasses, and tussocks | grass tussocks and moss                              | mix of grasses, mosses, and sedges                                 | grass tussocks
and mosses   |
| Organic matter content | 77.3%                               | 75.1%                                                | 71.8%                                                              | 45.5%                          |
| Unique characteristics | ephemeral, drainage
features     | homogeneous in
vegetation character
and height | located in a valley
bottom and partitioned
by Imnavait Creek | widespread
subsurface rocks |
| ALT                    | 49 cm                               | 39 cm                                                | 44 cm                                                              | 58 cm                          |

An improved presentation of results would be beneficial especially for comparison of ALT from field measurements and the modelled results. It is difficult for example, for the reader to compare the measured values to the ML outputs in figure 4 as the study plot area isn't clear. The text refers to ALT variability near water bodies, but the results presented (i.e. maps) do not allow the reader to see this. Consideration of the accuracy of the models in terms of the entire area covered is fine but it is useful to consider which areas of the study area and under what conditions the accuracy is better. (see below for additional comments)

**Response 1.3**

As suggested, Figure 4 was re-plotted to ensure ALT measurements and predictions were only mapped over the same overlapping areas for improved presentation. We also added the following in Section 4.1.1 for addressing the model performance and uncertainties.

"Considering the larger ALT variability and more diversified surface conditions at 0.1-m resolution relative to the 5-m resolution, the RF model trained using limited 5-m data set may not be able to fully capture the 0.1-m ALT variations, leading to inconsistency of the overall ALT patterns between the 5-m and 0.1-m results (e.g., Fig. 4d, 4e, 4f) and additional uncertainties in the multi-resolution analysis. In addition, the RF predictions tended to be centralized, which may underestimate larger ALT values and overestimate smaller ones relative to the measurements (e.g., Fig. 4g, 4h, and 4i)".

Figure 4: Comparisons of ALT spatial patterns derived from field measurements (a, d, g, j), 5-m machine learning outputs (b, e, h, k), and 0.1-m machine learning estimates (c, f, i, l) for the intensively sampled plots (grey shading indicates areas without sampling). The map was plotted using Canada Albers Equal Area Conic projection.

In the discussion and conclusions, general statements are made but it is unclear how the results and analysis support these statements. There needs to be a clearer link between results and conclusions.

**Response 1.4**

Thanks for the comment. We revised Section 5.1 to provide more in-depth analysis for interpreting our ALT observations and model analysis. The revisions are presented below.

"Consistent with previous studies, our field observations and RF estimates (section 4.1.1) confirmed that greater ALT is most common in areas with standing water or adjacent to creeks (e.g., Plot 5; Fig. 4g, 4h, 4i), where wet conditions enhance soil thermal conductivity in foothills tundra (Grant et al., 2017; Clayton et al., 2021) despite increased latent heat required for thawing. Relatively larger ALT was also found in the vicinity of subsurface rocks (e.g., within 1m distance) (e.g., Plot 6; Fig. 5j, 5k, 5l), whose high thermal conductivities facilitate heat propagation and summer thawing (Bonnaventure et al., 2013). Relatively lower ALT was recorded under shrubs, which likely cool the ground in summer through canopy shading (Lawrence and Swenson, 2011) and in winter through the thermal bridging effect (Domine et al., 2022). However, shrubs can also have a counteracting influence on ALT by promoting snow accumulation; whereby, the deeper snow layer insulates the ground, leading to warmer winter soil temperatures (e.g., Palmer et al., 2012; Morse et al., 2012; Kropp et al., 2020) which can result in a deeper active layer when this winter warming effect outweighs the summer shading effect of the shrubs (Way and Lapalme, 2021). A thick moss layer may also slow active layer thaw through its insulating capacity (Schuuring et al., 2024), though no specific descriptions of the moss layer were made in our sampling.

In the Imnavait Creek area, the thickness of the organic layer increases from hill crests to foot slopes. The thicker organic layer provides enhanced thermal insulation, leading to a shallower active layer downhill (Walker and Walker, 1996). Accordingly, Plots 3 and 4 on west-facing downhill slopes and Plot 5 in the valley bottom exhibited overall high soil organic matter content (~75%) and relatively low ALT (~44 cm). In contrast, the higher-elevation Plot 6 had lower organic matter (~46%) and a higher ALT (~58 cm). Besides terrain-controlled organic matter distribution, topography also affects ALT through its impacts on runoff and drainage, soil temperature, snow properties, and vegetation types (Walker and Walker, 1996, Li et al., 2017). Accordingly, topography information including slope (17.40%) and aspect (13.44%) are among the most important factors shaping ALT variations in the ML- and drone-based analysis, while surface features including vegetation, water bodies, and soil properties determined from the multi-spectral reflectance and optical-NIR indices, all showed important contributions to the 5-m ALT predictions over the sampling plots (7.24% to 14.05%; section 4.1.1).

It is noted that additional radar (L-band 1.26 GHz) observations from UAVSAR helped to enhance the performance of the first RF model (e.g., R increased from 0.78 to 0.81), but were not used in the subsequent scaling analysis (section 3.1.2). For the features selected for radar-based ALT predictions, red-edge reflectance (16.67%), HV-polarized radar backscatter (16.04%), aspect (15.33%), and HH-polarized radar backscatter (15.04%) contributed most to the predictions, while the red band (13.34%), slope (11.99%), and green band (11.57%) observations were relatively less important. The sensitivity of radar backscatter to vegetation biomass, surface water bodies, and soil wetness likely enhanced the ALT estimation.

For the ML- and satellite-based analysis, terrain factors (elevation, slope, and aspect) collectively dominate the ALT predictions (64.89% contribution) at 10-m resolution. The broad ALT patterns over the surrounding region (Fig. 5a) largely align with terrain-driven variability (section 4.1.2), as also observed in a previous study (Hinkel and Nelson, 2003). In general, south-facing slopes in the Northern Hemisphere receive more solar radiation than north-facing slopes, leading to warmer soil and larger ALT. However, the study region is characterized by gentle terrain slopes and west facing aspects (Fig. 5). Besides the terrain-controlled organic matter distribution observed over the Imnavait Creek area, direct solar radiation loading is higher around the hill tops and lower in downslope areas (Hinkel and Nelson, 2003), thus promoting larger ALT conditions in the uplands (Fig. 5). Topography therefore exerts a direct influence on the general thaw pattern as shown in the regional ML analysis".

**Added references:**

Walker, D. A. and Walker, M. D.: Terrain and vegetation of the Imnavait Creek watershed, Landscape function and disturbance in Arctic Tundra, pp. 73-108, Berlin, Heidelberg: Springer Berlin Heidelberg, 1996.

Li, A., Tan, X., Wu, W., Liu, H. and Zhu, J.: Predicting active-layer soil thickness using topographic variables at a small watershed scale, Plos one, 12(9), p.e0183742, 2017.

**Comment 1.5**

The organization of some sections of the manuscript could be improved including Section 2 and 4.1 – see further comments below. Tables could also be considered for summarizing ALT conditions for study plots etc.

**Response 1.5**

As suggested, Section 2 was revised for improving the structure, and providing a general description for the study region and more details for the sampling plot conditions. Table 1 summarizing the plot surface and ALT conditions was also added. Section 4.1.1 was re-organized to present results for each plot first before giving overall statistics. We also described the field sampling results first before comparing them with the RF model estimates.

The authors appear to confuse scale and resolution. Scale and resolution are not the same thing. The authors refer to scale (e.g. finer, coarser scales) in the manuscript, but this is incorrect, and references should be made to resolution.

**Response 1.6**

As suggested, we used "resolution" instead of "scale" in the revision throughout the manuscript.

**Comment 1.7**

Editorial revisions have been suggested to improve clarity. Additional comments for the author's consideration are provided below.

**Response 1.7**

We thank the reviewer for the constructive editorial revisions. We have implemented all suggested changes to enhance the manuscript's readability.

**Comment 1.8**

**Additional Comments**

Title – It might be sufficient to have a shorter title: "....over Arctic-foothills tundra, North Slope Alaska" and delete the last part. I think you should mention the location as the study is specific to a region.

**Response 1.8**

As suggested, the title was shortened as "Assessing spatial heterogeneity of active layer thickness over Arctic-foothills tundra, North Slope Alaska"

**Comment 1.9**

L16 – Revision suggested for first sentence" Changes in active layer thickness are used as an indicator of permafrost degradation" (I don't think the ECV part is necessary).

**Response 1.9**

As suggested, the first sentence was revised as "Changes in active layer thickness (ALT) are used as an indicator of permafrost degradation".

L17-18 – A thickness doesn't deepen, revise to: "Increases in ALT can...." Infrastructure damage results from ground instability so maybe that should be mentioned as changes in ALT do not directly cause infrastructure damage.

**Response 1.10**

As suggested, the sentence was revised as "Increases in ALT can lead to increased greenhouse gas emissions, altered hydrology and ecology, ground instability, and ..."

**Comment 1.11**

L33 – It should be clear that you are referring to atmospheric warming here rather than permafrost warming.

**Response 1.11:**

The reviewer is correct. We revise the sentence as "Permafrost in the northern high latitudes is undergoing rapid changes driven by enhanced atmospheric warming at roughly four times..."

**Comment 1.12**

L34-37 – Consider revising the sentence. Some of these things are a result of permafrost degradation while other things mentioned may promote it. Deepening of a layer doesn't sound right so refer to thicker active layer.

**Response 1.12:** The sentence was re-written for improved clarity as below.

"Complex environmental changes that accompany degrading permafrost include widespread earlier spring thawing and lengthening of the thaw season, shifts in seasonal snow cover properties, contrasting wetting and drying patterns, vegetation greening and browning, and increasing disturbances. Permafrost degradation, which involves active layer (layer on top of permafrost that undergoes seasonal freeze/thaw) thickening, may lead to ground surface deformation".

**Comment 1.13**

L40 – Biskaborn et al. (2019) was not about GHG emissions, so I suggest you delete it. You could also consider citing the review paper of Miner et al. (2022).

**Response 1.13:** Thanks for the suggestion. We removed the reference "Biskaborn et al. 2019" and cited "Miner et al., 2022".

**Added Reference:**

Miner, K. R., Turetsky, M. R., Malina, E., Bartsch, A., Tamminen, J., McGuire, A. D., Fix, A., Sweeney, C., Elder, C. D., and Miller, C. E.: Permafrost carbon emissions in a changing Arctic, Nature Reviews Earth & Environment, 3, 55-67, doi:10.1038/s43017-021-00230-3, 2022.

**Comment 1.14**

L41-42 – The accepted definition of the active layer comes from the IPA glossary (van Everdingen et al. 1998) so that should probably be cited. Establishment of active layer thickness as an essential climate variable is described in Smith and Brown (2009) so this is probably better reference than what you use.

**Response 1.14:** We followed the reviewer suggestion and revised the sentence as below.

"Active layer thickness (ALT), defined as "the thickness of the layer of the ground that is subject to annual thawing and freeing in areas underlain by permafrost" (Van Everdingen et al. 1998), is an essential climate variable for monitoring permafrost degradation (Smith and Brown 2009)".

**Added reference:**

Van Everdingen, R.O. ed.: Multi-language glossary of permafrost and related ground-ice terms in Chinese, English, French, German, Icelandic, Italian, Norwegian, polish, Romanian, Russian, Spanish, and Swedish. International Permafrost Association, Terminology Working Group, 1998.

Smith, S. and Brown, J.: Permafrost: permafrost and seasonally frozen ground, T7. Global Terrestrial Observing System GTOS 62, Food and Agriculture Organization of the United Nations (FAO), Rome, 2009.

**Comment 1.15**

L44 – You could just refer to spatial and temporal variability.

**Response 1.15:** As suggested, the sentence was revised as "Accurate mapping of ALT spatial and temporal variability is critical for understanding impacts of climate change..."

**Comment 1.16**

L47-49 – You are essentially saying that local microclimate is important.

**Response 1.16:** The reviewer's summary is correct. However, we prefer keeping the original sentence and providing readers more detailed descriptions of the factors affecting local scale ALT heterogeneity.

L55 – Are you referring to process models that determine ALT here? – Obu et al. (2019) model simulates TTOP not ALT.

**Response 1.17:** To be more accurate, we removed the reference Obu et al. 2019.

**Comment 1.18**

L61-66 – ALT is inferred from information acquired using these techniques. In the case of geophysical techniques, there can be other factors that result in similar signals. Most geophysical techniques are used to determine frozen/unfrozen interfaces, but this also requires knowledge of geology etc. to make the interpretations. Techniques like InSAR are used to determine changes in surface elevation but freezing and thawing are not the only reason movements of the ground occur. Other remote sensing techniques provide information that is used with thermal models to simulate ALT etc. It is difficult to say that any of these techniques are direct measures of ALT, and you should probably say that they are used to infer or provide data for models to estimate ALT.

**Response 1.18:** Thanks for the comments! Accordingly, we added the sentence below for a more accurate summary of the remote sensing techniques in estimating ALT.

"In sum, the remote sensing techniques provide information necessary to infer or model ALT."

**Comment 1.19**

L75-77 - Inferred through modelling?

**Response 1.19:** To be clearer, the sentence was re-written as below:

"Considering its dependence on surface conditions (Kelley et al., 2004), ALT can also be indirectly inferred from optical vegetation observations and relatively high-frequency radar backscatter signals using regression analysis (Gangodagamage et al., 2014; Widhalm et al., 2017)."

L85 – Do you mean "characterize" or "assess" rather than clarify? I think other studies have determined the various controls. Are you investigating the relevant importance of these?

**Response 1.20:** We revised the sentence as "...assess the underlying environmental controls on ALT patterns manifesting at different resolutions...". This was done by analyzing the relative importance of the RF predictors and the in-situ ALT measurements.

**Comment 1.21**

L87-115 – Study Region section. Normally this would include a general description of the regional setting – climate, geology, vegetation etc. and then details of the study plots would be provided. This section could benefit from better organization.

**Response 1.21:**

Thanks for the suggestion. We revised the Section 2 Study Region section to include a generate description of the surrounding region, and more detailed information for the study plots. The revisions are also presented below.

"Our study focused on the Imnavait Creek (68.6167° N, 149.3167° W) area within the Alaskan North Slope tundra foothills region (Fig. 1). The study region experiences a cold climate with a mean annual temperature of -7.4°C, ranging from an average of -17°C in January to 9.4°C in July (Schramm et al., 2007). Annual precipitation averages 340 mm, two-thirds of which falls as light rainfall during the summer (Schramm et al., 2007). The dominant vegetation consists of water-tolerant plants like tussock sedges and mosses, grasses and low shrubs (Schramm et al., 2007). Vegetation and soil patterns vary with slope, aspect, and drainage conditions, exhibiting a patchy distribution across the study domain (Hinkel and Nelson, 2003). The region is also characterized by diverse glacial landforms, including deposits, stream networks, and bedrock outcrops (Hinkel and Nelson, 2003). The region has gently rolling terrain, with elevations ranging from ~750 m to 980 m. Vegetation and soil patterns vary with slope, aspect, and drainage conditions, exhibiting a patchy distribution across the study domain (Hinkel and Nelson, 2003). The region is also characterized by diverse glacial landforms, including deposits, stream networks, and bedrock outcrops (Hinkel and Nelson, 2003). Increasing ALT was observed from both in-situ measurements at the Circumpolar Active Layer Monitoring (CALM) sites (https://www2.gwu.edu/~calm/data/north.htm) and regional ALT records (Liu et al., 2024).

The local study area consisted of four intensively sampled field plots (Plot 3, Plot 4, Plot 5, and Plot 6 in Fig. 1; 90m ×90m each) distributed across an elevation gradient along west facing hill slopes, and surrounded by a larger (5 km by 5 km) landscape domain used for analyzing ALT across different resolutions (Fig.1; Table 1). The field plots are dominated by tussock-forming sedges and moss-lichen mats, along with scattered dwarf birch shrubs, and riparian grasses (Fig.

1-2). An informal survey of the common species identified dwarf birch (*Betula nana*), alpine blueberry (*Vaccinium uliginosum*), black bearberry (*Arctous* spp.), crowberry (*Empetrum nigrum*), and *Arctostaphylos uva-ursi*, among other less-recognizable species. Our soil coring samples indicated high organic matter content of the topsoil (0-10 cm depth) for Plots 3, 4 and 5, with the respective values of 77.3%, 75.1%, and 71.8%, contrasting with a relatively low value of 45.5% for Plot 6.

Plot 3 and Plot 4 are located on west-facing downhill slopes characterized by gradual elevation changes (10-15 m across the plot), small water tracks, and scattered glacial erratics. Plot 3 is crossed by multiple drainage features that are transverse to the Kuparuk River (Fig. 1). These features are evident in the topography as well as the vegetation, with alternating bands of dwarf willow, grasses, and tussocks. Standing water is present in many places, as are glacial erratics, particularly along the drainage channels. Plot 4 is more homogeneous in vegetation character and height, consisting mostly of grass tussocks and moss, with some larger glacial erratics present.

Plots 5 and 6 are on the east side of the access road and are more level than Plots 3 and 4. Plot 5 is located at a valley bottom and is partitioned by Imnavait Creek, which winds directly through the middle of the plot. Shallow drainage channels transverse to Imnavait Creek are found on the west side of the plot, while the east side is upland and drier. Vegetation cover in Plot 5 follows the typical mix of grasses, mosses, and sedges, with some moss forming humps about 20 cm tall, while shrub density and height increase close to the creek. Along Imnavait Creek there are taller (0.5-1.0 m) birch shrubs and more grasses. Plot 6 is characterized by widespread subsurface rocks (Fig. 1) and overall short-statured grass-sedge tussocks and mosses, lichens, small bunchgrasses, *Arctous spp.*, and alpine blueberry, interrupted by areas of bare rock and gravel. Bare areas are common in the center of the plot".

**Comment 1.22**

L88-89 – Normally a more general description of regional climate would be presented first, and this sentence seems out of place (see previous comment). Provide reference period for statements such as this and be clear that it is air temperature (rather than permafrost temperature) that is rising by the amount indicated.

**Response 1.22:** To be clearer, the sentence was deleted in the revision.

**Comment 1.23**

L100 – information on map projections should be provided in the figure caption.

**Response 1.23:** As suggested, we deleted the sentence and provided the map projection information in the figure caption.

L105 – Figure 1 – If the images include the plot area, then it should be clear what area of them is covered by the plot. The orientation of the plots and images differ so it is difficult for the reader to see the characteristics of the plots. It is also unclear if the scale of map and the images is the same.

**Response:** Figure 1 was re-plotted to show the same overlapping areas of the sampling plots and the corresponding drone images. We also clarified in the figure caption that the drone images did not use map scales and were for visual inspection only.

Figure 1: The study region encompasses an Arctic tundra area (68.6167°, -149.3167°; red dot in the inset) in the northern foothills of the Brooks Range, Alaska. The region consists of four intensively sampled plots (Plot 3, Plot 4, Plot 5, and Plot 6; 90 m x 90 m each) with their corresponding true-color RGB (red-green-blue) drone images displayed alongside for visual inspection only, and surrounded by a larger 5 km by 5 km study region (red rectangle) used for analyzing ALT scaling effects. The map was plotted using Canada Albers Equal Area Conic projection.

L125-126 – It would be better to indicate that mechanical probing to the depth of refusal was used to determine the depth of the frost table. Essentially that is what is done when probing is conducted.

**Response 1.25:** Thanks for the suggestion. We added the sentence below accordingly.

"The mechanical probing to the depth of refusal was used to determine the depth of the frost table".

**Comment 1.26**

L129-131 – Revision suggested: "Additional observations were made including vegetation type and distribution, occurrence of standing and running water...." Observation of subsurface rocks is mentioned but were there also descriptions of organic layer thickness or surficial materials which are relevant for interpretation of results.

**Response 1.26:** As suggested, the sentence was revised as "Additional observations were made including vegetation types, occurrence of standing and running water, and appearance of subsurface rocks".

In addition, we also took soil core samples for quantifying organic layer for the sampling plots. The following sentence was added:

"Soil organic layer properties (section 2) were also estimated from soil cores taken within the sampling plots".

**Comment 1.27**

L169-170 – The way this is written it sounds like ALT is directly determined through remote sensing. Don't you mean that the RF method is used with parameters determined through remote sensing to estimate ALT. Doesn't the ALT used to develop the model come from the field observations?

**Response 1.27:** To be more accurate, the sentence was re-written as below.

"The RF method has been widely used with parameters determined through remote sensing to estimate land parameters such as soil moisture, vegetation optical depth, and ALT...".

L224-255 Section 4.1.1 – It would be better to present results for each plot first and then compare them before giving overall statistics. Clearly there are differences between the plots, and they should be described first. A better presentation of the results of the field sampling should be provided before presenting results of the RF models and the comparison to observed ALT.

**Response 1.28:** As suggested, section 4.1.1 was re-organized to present results for each plot first before giving overall statistics. In addition, field sampling results were presented before comparing with RF model estimates. The revised section was also given below.

"The ALT values sampled across all plots have a mean of 45.9 cm and a standard deviation of 11.9 cm. Relatively low ALT (39.3 cm) was typically found in areas dominated by non-riparian shrubs, while larger ALT occurred in soils near standing water (61.0 cm), along creeks (78.2 cm) (e.g., Fig. 4g), and around rocks (70.0 cm) (e.g., Fig. 4j).

The resulting 5-m and 0.1-m ALT maps were compared with the field measurements (Fig. 4). The 5-m ALT maps (Fig. 4b, 4e, 4h, 4k) captured the primary ALT patterns observed from the field measurements (Fig. 4a, 4d, 4g, 4j) including elevated ALT along the perimeter of Plot 3, consistently low ALT throughout Plot 4, high ALT values following the water tracks in Plot 5, and generally deep ALT in Plot 6. The 0.1-m results revealed significantly finer variations (Fig. 4c, 4f, 4i, and 4l), resolving ALT patterns associated with small water tracks (e.g., Fig. 4i), discrete vegetation patches (Fig. 4c, 4l), and widespread clusters of underlying rocks (Fig. 4l) that were not discernible at 5-m resolution.

Considering the larger ALT variability and more diversified surface conditions at 0.1-m resolution relative to the 5-m resolution, the RF model trained using limited 5-m data set may not be able to fully capture the 0.1-m ALT variations, leading to inconsistency of the overall ALT patterns between the 5-m and 0.1-m results (e.g., Fig. 4d, 4e, 4f) and additional uncertainties in the multi-resolution analysis. In addition, the RF predictions tended to be centralized, which may underestimate larger ALT values and overestimate smaller ones relative to the measurements (e.g., Fig. 4g, 4h, and 4i).

Overall, the first RF model was able to reproduce the sampled ALT at 5-m resolution with a RMSE of 6.53 cm and strong correlation (0.78). Among all the predictors, slope (17.40%), rededge reflectance (14.05%), aspect (13.44%), and NIR reflectance (12.64%) were the most important features contributing to ALT predictions. The contributions from other predictors including green band (9.91%), blue band (9.43%), NDWI (8.48%), red band (7.41%), and NDVI (7.24%) were also important. The RF model applied to the 0.1-m predictor features was able to capture the high-resolution ALT variability and details missed by coarser-resolution results".

L236-240 – These features do not appear to be visible on the maps in Figure 4 so difficult for the reader to see how you arrive at these interpretations. There appears to be substantial difference between Plot 5 (g) observed ALT and modelled (e) – observed values appear to be less than modelled.

**Response 1.29:**

For better comparisons between the 5-m and 0.1-m results, Figure 4 was re-plotted and additional explanations were provided to address the discrepancies between the ALT results and possible uncertainties as below.

"...The 0.1-m results revealed significantly finer variations (Fig. 4c, 4f, 4i, and 4l), resolving ALT patterns associated with small water tracks (e.g., Fig. 4i), discrete vegetation patches (Fig. 4c, 4l), and widespread clusters of underlying rocks (Fig. 4l) that were not discernible at 5-m resolution.

Considering the larger ALT variability and more diversified surface conditions at 0.1-m resolution relative to the 5-m resolution, the RF model trained using limited 5-m data set may not be able to fully capture the 0.1-m ALT variations, leading to inconsistency of the overall ALT patterns between the 5-m and 0.1-m results (e.g., Fig. 4d, 4e, 4f) and additional uncertainties in the multi-resolution analysis. In addition, the RF predictions tended to be centralized, which may underestimate larger ALT values and overestimate smaller ones relative to the measurements (e.g., Fig. 4g, 4h, and 4i)".

Figure 4: Comparisons of ALT spatial patterns derived from field measurements (a, d, g, j), 5-m machine learning outputs (b, e, h, k), and 0.1-m machine learning estimates (c, f, i, l) for the intensively sampled plots (grey shading indicates areas without sampling). The map was plotted using Canada Albers Equal Area Conic projection.

Figure 4 – The presentation does not allow the reader to compare the observed to the model outputs as it is unclear how the plot area in first column fits on the maps in the other two columns. The plot area should be clearly shown on the other plots. For plot 6 the rest of the plot area should be shown in (j) with grey shading for example to indicate area that couldn't be probed.

**Response 1.30:** As suggested, Figure 4 was re-plotted for improved comparisons between the ALT results. In addition, grey shading was added to indicate areas without sampling.

Figure 4: Comparisons of ALT spatial patterns derived from field measurements (a, d, g, j), 5-m machine learning outputs (b, e, h, k), and 0.1-m machine learning estimates (c, f, i, l) for the intensively sampled plots (grey shading indicates areas without sampling). The map was plotted using Canada Albers Equal Area Conic projection.

L245-255 – We would expect warmer conditions and greater ALT on south facing vs north facing slopes – can you say anything about this based on observed results. Note that some of the factors considered are related. For example, vegetation will depend on elevation and aspect. Drainage and therefore surface wetness (affects vegetation) will depend on topography.

**Response 1.31:** We expanded the discussion to address the ALT control from topography and its interactions with other factors as below.

"In the Imnavait Creek area, the thickness of the organic layer increases from hill crests to foot slopes. The thicker organic layer provides enhanced thermal insulation, leading to a shallower active layer downhill (Walker and Walker, 1996). Accordingly, Plots 3 and 4 on west-facing downhill slopes and Plot 5 in the valley bottom exhibited overall high soil organic matter content (~75%) and relatively low ALT (~44 cm). In contrast, the higher-elevation Plot 6 had lower organic matter (~46%) and a higher ALT (~58 cm). Besides terrain-controlled organic matter distribution, topography also affects ALT through its impacts on runoff and drainage, soil temperature, snow properties, and vegetation types (Walker and Walker, 1996, Li et al., 2017). Accordingly, topography information including slope (17.40%) and aspect (13.44%) are among the most important factors shaping the ALT variations in the ML- and drone-based analysis, while the surface features including vegetation, water bodies, and soil properties determined from the multi-spectral reflectance and optical-NIR indices, all showed important contributions to the 5-m ALT predictions over the sampling plots (7.24% to 14.05%; section 4.1.1).

. . .

For the ML- and satellite-based analysis, terrain factors (elevation, slope, and aspect) collectively dominate the ALT predictions (64.89% contribution) at 10-m resolution. The broad ALT patterns over the surrounding region (Fig. 5a) largely align with terrain-driven variability (section 4.1.2), as also observed in a previous study (Hinkel and Nelson, 2003). In general, south-facing slopes in the Northern Hemisphere receive more solar radiation than north-facing slopes, leading to warmer soil and larger ALT. However, the study region is characterized by gentle terrain slopes and west facing aspects (Fig. 5). Besides the terrain-controlled organic matter distribution observed over the Imnavait Creek area, direct solar radiation loading is higher around the hill tops and lower in downslope areas (Hinkel and Nelson, 2003), thus promoting larger ALT conditions in the uplands (Fig. 5). Topography therefore exerts a direct influence on the general thaw pattern as shown in the regional ML analysis".

L262-264 – Words like "rapid" and "slower" imply that a change over time is being considered but that is not the case here. It would be better to refer to is a smaller increase in uncertainty at coarser (or lower) resolution (note on you map in Figure 4 the x axis appears to be the resolution, not scale – resolution and scale are not the same thing).

**Response 1.32:** As suggested, the sentence was re-written to be more rigorous as below. In addition, the X-axis label was revised as "Spatial resolution (m)".

"Uncertainties increase as spatial resolution becomes coarser. However, this increase is most pronounced when moving from high resolutions (e.g., sub-meter to 1 m) and becomes smaller at coarser pixel sizes (e.g., 100 to 1000 m)".

**Comment 1.33**

L276-278 – "Coarser scale" is incorrect, it should be "coarser resolution". Air temperature affects surface temperature (as do local environmental factors that affect microclimate) which influence the ground thermal regime (ground temperature) and therefore active layer conditions.

**Response 1.33:** Thanks for the interpretation and correction. We used "resolution" instead of "scale", and expanded the sentence by adopting the reviewer's interpretation. The revised sentences are given below.

"Regional ALT dynamics at kilometer or coarser resolutions are primarily governed by air temperature, which affects surface temperature, and further influences soil thermal regimes and therefore active layer conditions (Gangodagamage et al., 2014; Peng et al., 2023). Local ALT variations at finer resolutions (sub-meter to 100 m) are highly complex (Gangodagamage et al., 2014) since soil thermal regimes are further modified by local environmental factors that fine-tune the microclimate".

**Comment 1.34**

L275-302 – Section 5.1 – There are a lot of general statements from the literature, but very little analysis is presented to show the relative importance of the various factors mentioned. Information on snow cover, soil texture, groundwater flow etc. has not been presented and it is unclear how these things may vary over the study area. You mention that ALT is greater in areas with standing water or adjacent to creeks but not clear from results presented (e.g. maps) that this is the case.

**Response:** We revised Section 5.1 to provide more in-depth analysis to interpret our ALT observations and model analysis, and clearer links between the interpretation and our observations/predictions. The revisions are presented below.

"Consistent with previous studies, our field observations and RF estimates (section 4.1.1) confirmed that greater ALT is most common in areas with standing water or adjacent to creeks (e.g., Plot 5; Fig. 4g, 4h, 4i), where wet conditions enhance soil thermal conductivity in foothills tundra (Grant et al., 2017; Clayton et al., 2021) despite increased latent heat required for thawing. Relatively larger ALT was also found in the vicinity of subsurface rocks (e.g., within 1m distance) (e.g., Plot 6; Fig. 5j, 5k, 5l), whose high thermal conductivities facilitate heat propagation and summer thawing (Bonnaventure et al., 2013). Relatively lower ALT was recorded under shrubs, which likely cool the ground in summer through canopy shading (Lawrence and Swenson, 2011) and in winter through the thermal bridging effect (Domine et al., 2022). However, shrubs can also have a counteracting influence on ALT by promoting snow accumulation; whereby, the deeper snow layer insulates the ground, leading to warmer winter soil temperatures (e.g., Palmer et al., 2012; Morse et al., 2012; Kropp et al., 2020) which can result in a deeper active layer when this winter warming effect outweighs the summer shading effect of the shrubs (Way and Lapalme, 2021). A thick moss layer may also slow active layer thaw through its insulating capacity (Schuuring et al., 2024), though no specific descriptions of the moss layer were made in our sampling.

In the Imnavait Creek area, the thickness of the organic layer increases from hill crests to foot slopes. The thicker organic layer provides enhanced thermal insulation, leading to a shallower active layer downhill (Walker and Walker, 1996). Accordingly, Plots 3 and 4 on west-facing downhill slopes and Plot 5 in the valley bottom exhibited overall high soil organic matter content (~75%) and relatively low ALT (~44 cm). In contrast, the higher-elevation Plot 6 had lower organic matter (~46%) and a higher ALT (~58 cm). Besides terrain-controlled organic matter distribution, topography also affects ALT through its impacts on runoff and drainage, soil temperature, snow properties, and vegetation types (Walker and Walker, 1996, Li et al., 2017). Accordingly, topography information including slope (17.40%) and aspect (13.44%) are among the most important factors shaping ALT variations in the ML- and drone-based analysis, while surface features including vegetation, water bodies, and soil properties determined from the multi-spectral reflectance and optical-NIR indices, all showed important contributions to the 5-m ALT predictions over the sampling plots (7.24% to 14.05%; section 4.1.1).

It is noted that additional radar (L-band 1.26 GHz) observations from UAVSAR helped to enhance the performance of the first RF model (e.g., R increased from 0.78 to 0.81), but were not used in the subsequent scaling analysis (section 3.1.2). For the features selected for radar-based ALT predictions, red-edge reflectance (16.67%), HV-polarized radar backscatter (16.04%), aspect (15.33%), and HH-polarized radar backscatter (15.04%) contributed most to the predictions, while the red band (13.34%), slope (11.99%), and green band (11.57%) observations

were relatively less important. The sensitivity of radar backscatter to vegetation biomass, surface water bodies, and soil wetness likely enhanced the ALT estimation.

For the ML- and satellite-based analysis, terrain factors (elevation, slope, and aspect) collectively dominate the ALT predictions (64.89% contribution) at 10-m resolution. The broad ALT patterns over the surrounding region (Fig. 5a) largely align with terrain-driven variability (section 4.1.2), as also observed in a previous study (Hinkel and Nelson, 2003). In general, south-facing slopes in the Northern Hemisphere receive more solar radiation than north-facing slopes, leading to warmer soil and larger ALT. However, the study region is characterized by gentle terrain slopes and west facing aspects (Fig. 5). Besides the terrain-controlled organic matter distribution observed over the Imnavait Creek area, direct solar radiation loading is higher around the hill tops and lower in downslope areas (Hinkel and Nelson, 2003), thus promoting larger ALT conditions in the uplands (Fig. 5). Topography therefore exerts a direct influence on the general thaw pattern as shown in the regional ML analysis".

**Added reference:**

Walker, D. A. and Walker, M. D.: Terrain and vegetation of the Imnavait Creek watershed, Landscape function and disturbance in Arctic Tundra, pp. 73-108, Berlin, Heidelberg: Springer Berlin Heidelberg, 1996.

Li, A., Tan, X., Wu, W., Liu, H. and Zhu, J.: Predicting active-layer soil thickness using topographic variables at a small watershed scale, Plos one, 12(9), p.e0183742, 2017.

**Comment 1.35**

L282-285 – Revise "deeper ALT" to "greater ALT" (a thickness can't be deeper – same issue with shallow ALT). Latent heat is also an important factor with respect to the effect that wet conditions have on the ground thermal regime.

**Response 1.35**: As suggested, the sentence was revised as below.

"Consistent with previous studies, our field observations and RF estimates (section 4.1.1) confirmed that greater ALT is most common in areas with standing water or adjacent to creeks (e.g., Plot 5; Fig. 4g, 4h, 4i), where wet conditions enhance soil thermal conductivity in foothills tundra (Grant et al., 2017; Clayton et al., 2021) despite increased latent heat required for thawing. Relatively larger ALT was also found in the vicinity of subsurface rocks (e.g., within 1-m distance) (e.g., Plot 6; Fig. 5j, 5k, 5l), whose high thermal conductivities facilitate heat propagation and summer thawing (Bonnaventure et al., 2013)".

L287-290 – Note that shrubs can promote snow accumulation and other studies have shown that this leads to warmer winter ground temperatures (e.g. Palmer et al. 2012; Morse et al. 2012; Way and Lapalme 2021; Kropp et al. 2020) – winter conditions will influence ALT and it is not as simple as implied in the text. Way and Lapalme (2021) also showed that the insulating effect of snow outweighs the shading effect of shrubs.

**Response 1.36:** For a more rigorous analysis, we added the following discussions on the shrub impacts as below.

"Relatively lower ALT was recorded over the shrubs, which likely cool the ground in summer through canopy shading (Lawrence and Swenson, 2011) and in winter through thermal bridging effect (Domine et al., 2022). However, shrubs can also have a counteracting influence on ALT by promoting snow accumulation; whereby, the deeper snow layer insulates the ground, leading to warmer winter soil temperatures (e.g., Palmer et al., 2012; Morse et al., 2012; Kropp et al., 2020) which can result in a deeper active layer when this winter warming effect outweighs the summer shading effect of the shrubs (Way and Lapalme, 2021)".

**Added references:**

Kropp, H., Loranty, M. M., Natali, S. M., Kholodov, A. L., Rocha, A. V., Myers-Smith, I., Abbot, B. W., Abermann, J., Blanc-Betes, E., Blok, D. and Blume-Werry, G.: Shallow soils are warmer under trees and tall shrubs across Arctic and Boreal ecosystems, Environmental research letters, 16(1), p.015001, 2020.

Morse, P. D., Burn, C. R., and Kokelj, S. V.: Influence of snow on near-surface ground temperatures in upland and alluvial environments of the outer Mackenzie Delta, Northwest Territories, Canadian Journal Earth Sciences, 49: 895-913. doi:10.1139/E2012-012, 2012.

Palmer, M. J., Burn, C. R., and Kokelj, S. V.: Factors influencing permafrost temperatures across tree line in the uplands east of the Mackenzie Delta, 2004–2010, Canadian Journal of Earth Sciences, 49: 877-894. doi:10.1139/E2012-002, 2012.

Way, R. G., and Lapalme, C. M.: Does tall vegetation warm or cool the ground surface? Constraining the ground thermal impacts of upright vegetation in northern environments, Environmental Research Letters, 16: 054077. doi:10.1088/1748-9326/abef31, 2021

L306 – I think you mean greater uncertainty in ALT at sub-metre resolution.

**Response 1.37:** The ALT uncertainty increases with coarser spatial resolution, but the increase is greater at finer spatial resolutions. We revised the sentence for improved clarity as below.

"Our analysis revealed a greater increase in ALT spatial uncertainty at the sub-meter resolutions"

**Comment 1.38**

L322-323 – This is a general statement but not a conclusion of your study – there was no investigation of GHG emission, infrastructure issues etc.

**Response 1.38:** The sentence was deleted in the revision to avoid confusion.

**Comment 1.39**

L350 – References – check URL links numbers as some of them do not seem to work. I noticed this with a few ERL publications.

**Response 1.39:** Thanks for the careful check. All the URL links were checked and updated to make sure they are workable.

**Comment 1.40**

L599 – Biskaborn et al. has many more coauthors so "and coauthors" should be added after the last author given. Same comment for Obu et al. in line 457.

**Response 1.40:** The two references were not cited in the revised manuscript. For a few other references with many more coauthors, we added "and co-authors" as suggested.

**Comment 1.41**

References cited in comments

Kropp, H. et al., 2021. Shallow soils are warmer under trees and tall shrubs across Arctic and Boreal ecosystems. Environmental Research Letters, 16: 015001. doi: 10.1088/1748-9326/abc994

Miner, K.R., Turetsky, M.R., Malina, E., Bartsch, A., Tamminen, J., McGuire, A.D., Fix, A., Sweeney, C., Elder, C.D., and Miller, C.E. 2022. Permafrost carbon emissions in a changing Arctic. Nature Reviews Earth & Environment, 3: 55-67. doi:10.1038/s43017-021-00230-3

Morse, P.D., Burn, C.R., and Kokelj, S.V. 2012. Influence of snow on near-surface ground temperatures in upland and alluvial environments of the outer Mackenzie Delta, Northwest Territories. Canadian Journal Earth Sciences, 49: 895-913. doi:10.1139/E2012-012

Palmer, M.J., Burn, C.R., and Kokelj, S.V. 2012. Factors influencing permafrost temperatures across tree line in the uplands east of the Mackenzie Delta, 2004–2010. Canadian Journal of Earth Sciences, 49: 877-894. doi:10.1139/E2012-002

Smith, S. and Brown, J., 2009. Permafrost: permafrost and seasonally frozen ground, T7. Global Terrestrial Observing System GTOS 62, Food and Agriculture Organization of the United Nations (FAO), Rome.

Way, R.G., and Lapalme, C.M. 2021. Does tall vegetation warm or cool the ground surface? Constraining the ground thermal impacts of upright vegetation in northern environments. Environmental Research Letters, 16: 054077. doi:10.1088/1748-9326/abef31

**Response 1.41:** Thanks for summarizing the references, which were also cited in the revision to support the study.

---

## Author Comment (AC2)

RC2: 'Comment on egusphere-2025-3236', Anonymous Referee #2, 01 Sep 2025

**Comment 2.1**

The manuscript by Du et al. investigates active layer thickness (ALT) in the North Slope of Alaska using a combination of machine learning, intensive field sampling, and multi-source remote sensing data. The authors address ALT variability across multiple spatial scales. The main findings highlight the drivers of ALT at different scales. Additionally, the study quantifies scale-dependent uncertainties in ALT mapping and identifies functional relationships describing how these uncertainties change with spatial resolution. The paper is well-written and addresses an important topic in permafrost research. There are, however, some points that could be further discussed and a number of comments that should be addressed.

**Response 2.1:** Thanks for the careful review and insights into the study! We revised our manuscript accordingly. Please check our point-by-point responses below. All revisions were marked in blue.

**Comment 2.2**

The parameter importance at different resolutions is reported and supported by previous studies in the discussion section (L292-302). The discussion could be extended to provide more in-depth interpretation of the underlying mechanisms and implications.

**Response 2.2:** Thanks for the suggestion. We extended the discussion for more in-depth interpretation of the mechanisms underlying the ALT patterns described by in-situ observations and RF model predictions. The revisions are also presented below.

"Consistent with previous studies, our field observations and RF estimates (section 4.1.1) confirmed that greater ALT is most common in areas with standing water or adjacent to creeks (e.g., Plot 5; Fig. 4g, 4h, 4i), where wet conditions enhance soil thermal conductivity in foothills tundra (Grant et al., 2017; Clayton et al., 2021) despite increased latent heat required for thawing. Relatively larger ALT was also found in the vicinity of subsurface rocks (e.g., within 1-m distance) (e.g., Plot 6; Fig. 5j, 5k, 5l), whose high thermal conductivities facilitate heat propagation and summer thawing (Bonnaventure et al., 2013). Relatively lower ALT was recorded under shrubs, which likely cool the ground in summer through canopy shading (Lawrence and Swenson, 2011) and in winter through the thermal bridging effect (Domine et al., 2022). However, shrubs can also have a counteracting influence on ALT by promoting snow accumulation; whereby, the deeper snow layer insulates the ground, leading to warmer winter soil temperatures (e.g., Palmer et al., 2012; Morse et al., 2012; Kropp et al., 2020) which can result in a deeper active layer when this winter warming effect outweighs the summer shading effect of the shrubs (Way and Lapalme, 2021). A thick moss layer may also slow active layer

thaw through its insulating capacity (Schuuring et al., 2024), though no specific descriptions of the moss layer were made in our sampling.

In the Imnavait Creek area, the thickness of the organic layer increases from hill crests to foot slopes. The thicker organic layer provides enhanced thermal insulation, leading to a shallower active layer downhill (Walker and Walker, 1996). Accordingly, Plots 3 and 4 on west-facing downhill slopes and Plot 5 in the valley bottom exhibited overall high soil organic matter content (~75%) and relatively low ALT (~44 cm). In contrast, the higher-elevation Plot 6 had lower organic matter (~46%) and a higher ALT (~58 cm). Besides terrain-controlled organic matter distribution, topography also affects ALT through its impacts on runoff and drainage, soil temperature, snow properties, and vegetation types (Walker and Walker, 1996, Li et al., 2017). Accordingly, topography information including slope (17.40%) and aspect (13.44%) are among the most important factors shaping ALT variations in the ML- and drone-based analysis, while surface features including vegetation, water bodies, and soil properties determined from the multi-spectral reflectance and optical-NIR indices, all showed important contributions to the 5-m ALT predictions over the sampling plots (7.24% to 14.05%; section 4.1.1).

It is noted that additional radar (L-band 1.26 GHz) observations from UAVSAR helped to enhance the performance of the first RF model (e.g., R increased from 0.78 to 0.81), but were not used in the subsequent scaling analysis (section 3.1.2). For the features selected for radar-based ALT predictions, red-edge reflectance (16.67%), HV-polarized radar backscatter (16.04%), aspect (15.33%), and HH-polarized radar backscatter (15.04%) contributed most to the predictions, while the red band (13.34%), slope (11.99%), and green band (11.57%) observations were relatively less important. The sensitivity of radar backscatter to vegetation biomass, surface water bodies, and soil wetness likely enhanced the ALT estimation.

For the ML- and satellite-based analysis, terrain factors (elevation, slope, and aspect) collectively dominate the ALT predictions (64.89% contribution) at 10-m resolution. The broad ALT patterns over the surrounding region (Fig. 5a) largely align with terrain-driven variability (section 4.1.2), as also observed in a previous study (Hinkel and Nelson, 2003). In general, south-facing slopes in the Northern Hemisphere receive more solar radiation than north-facing slopes, leading to warmer soil and larger ALT. However, the study region is characterized by gentle terrain slopes and west facing aspects (Fig. 5). Besides the terrain-controlled organic matter distribution observed over the Imnavait Creek area, direct solar radiation loading is higher around the hill tops and lower in downslope areas (Hinkel and Nelson, 2003), thus promoting larger ALT conditions in the uplands (Fig. 5). Topography therefore exerts a direct influence on the general thaw pattern as shown in the regional ML analysis".

**Added reference:**

Walker, D. A. and Walker, M. D.: Terrain and vegetation of the Imnavait Creek watershed, Landscape function and disturbance in Arctic Tundra, pp. 73-108, Berlin, Heidelberg: Springer Berlin Heidelberg, 1996.

Li, A., Tan, X., Wu, W., Liu, H. and Zhu, J.: Predicting active-layer soil thickness using topographic variables at a small watershed scale, Plos one, 12(9), p.e0183742, 2017.

**Comment 2.3**

Using the 0.1 m results as a benchmark raises some questions. In Figure 4 (especially panel f), these values appear to differ from both in situ measurements and the 5 m results, which could be further discussed. Since the 0.1 m data are synthetically generated for the scaling effects analysis, it would be helpful if the authors could comment on any potential influence on the results.

**Response 2.3:** We agree with the reviewer and added the following in section 4.1.1 to address the uncertainties:

"Considering the larger ALT variability and more diversified surface conditions at 0.1-m resolution relative to the 5-m resolution, the RF model trained using limited 5-m data set may not be able to fully capture the 0.1-m ALT variations, leading to inconsistency of the overall ALT patterns between the 5-m and 0.1-m results (e.g., Fig. 4d, 4e, 4f) and additional uncertainties in the multi-resolution analysis. In addition, the RF predictions tended to be centralized, which may underestimate larger ALT values and overestimate smaller ones relative to the measurements (e.g., Fig. 4g, 4h, and 4i)".

**Comment 2.4**

Furthermore, the set of predictors chosen for the 5 m and 10 m models are not the same, with many variables omitted from the 10 m model. Could the authors clarify the rationale behind this choice? Since red-edge reflectance has relatively high importance at 5 m, it could have been valuable to investigate the 10 m red-edge from Sentinel-2 by performing super-resolution on the 20 m red-edge band. Additionally, elevation, which appears to have the highest importance for the 10 m model, was not included for the higher-resolution 5 m model. Could the authors comment on these choices and their potential implications?

**Response 2.4:** Thanks for the comment. The predictors were selected based on the "leave one feature out" (LOFO) approach (Liu et al., 2013). If removing a feature led to improved or unchanged RF performance, the feature was not selected since it was less important or redundant. For example, for the 5-m RF model, removing elevation feature led to essentially no performance change (R change less than 0.01; RMSE change less than 0.25 cm). This indicates that terrain control on 5-m ALT distributions over the sampling plots can be represented by aspect and slope. Similarly, the removing of Sentinel-2 red-edge bands reflectance led to negligible performance change for the 10-m RF model, which is likely due to the dominant control from terrain factors on the ALT distributions at 10-m resolution.

Reference:

Liu, J., Danait, N., Hu, S. and Sengupta, S., 2013, December. A leave-one-feature-out wrapper method for feature selection in data classification. In 2013 6th International Conference on Biomedical Engineering and Informatics (pp. 656-660). IEEE.

**Comment 2.5**

For the machine learning predictors at 5 m resolution, the DEM derived from RGB imagery was used. Were any differences investigated between this DEM and the ArcticDEM aggregated to the same 5 m resolution? Could the authors discuss any potential consequences of using the RGB-derived DEM compared to ArcticDEM?

**Response 2.5:** As suggested, we compared the two DEM data sets at the same 5-m resolution. The two DEMs were highly correlated (R = 0.99) across the four plots, though ArcticDEM is consistently higher in elevation for approximately 5.1 m than the drone-based DEM. After removing the bias, the RMSE between the two DEMs is 0.47m. The associated 5-m ALT predictions using ArcticDEM performed nearly as well as those from the drone-based DEM, with RMSE increasing from 6.53 cm to 7.24 cm and correlation declining slightly from 0.78 to 0.72.

**Comment 2.6**

The study focuses on a specific region. Could you comment on the transferability of the model to other regions? Would additional in situ training data be required for applying the approach elsewhere? Are similar parameter importances and scaling effects to be expected in other regions, and how representative is the study area for the broader Arctic context?

**Response 2.6:** Thanks for the comments!**

Based on the Circumpolar Arctic Vegetation Map (CAVM) (Raynolds et al., 2019), plot 3 and 4 belong to vegetation type G3 (Non-tussock sedge, dwarf-shrub, moss tundra) while plot 5 and 6 belong to vegetation type S1 (Erect dwarf-shrub, moss tundra), where G3 and S1 together represent 29.8% area of the Arctic (excluding glacier and water bodies). The Arctic here was defined as the "area of the Earth with tundra vegetation, an arctic climate and arctic flora, with the tree line defining the southern limit" (Raynolds et al., 2019). Accordingly, the model, parameter importance and scaling effects are more suitable for representing the Arctic G3 and S1 areas relative to other regions. However, the study area is also unique in its abundance of glacier deposits and low rolling terrain, and the in-situ measurements were only from one season. For a more rigorous assessment of the applicability or transferability of the data-driven model for other regions, multiple-season measurements over a few spatially-distributed regions within the Arctic would be necessary.

Accordingly, we added the following in the discussion:

"The model and analysis from this study are more applicable to the Arctic region with similar vegetation covers (~29.8% area of the Arctic excluding glacier and water bodies; Raynolds et al., 2019). However, considering the limited sampling areas and time period, multiple-season measurements over spatially-distributed regions within the Arctic would be necessary for extending the data-driven study in its spatial and temporal representativeness".

**Added reference:**

Raynolds, M. K., Walker, D. A., Balser, A., Bay, C., Campbell, M., Cherosov, M. M., Daniëls, F. J., Eidesen, P. B., Ermokhina, K. A., Frost, G. V. and Jedrzejek, B.: A raster version of the Circumpolar Arctic Vegetation Map (CAVM), Remote Sensing of Environment, 232, p.111297, https://doi.org/10.1016/j.rse.2019.111297, 2019.

**Comment 2.7**

You are mentioning the importance of not only ALT spatial distributions but also temporal dynamics (L44). Could you maybe comment on the feasibility of resolving ALT temporal dynamics with the current approach? For instance, what is the potential for detecting interannual variations in ALT with the presented approach? In addition, since the Sentinel-2 data originate from multiple summer months (L151), could the authors comment on how intra-seasonal variability might influence the results?

**Response 2.7:** Thanks for the comment! The ALT inter-annual variations are affected by many dynamic factors (e.g., air temperature, snow cover properties, and disturbances), which are not explicitly accounted for by the current model. Multi-season ALT measurements and inclusion of temporally variant predictors in the data-driven model would help better capture the ALT interannual dynamics. Accordingly, we added the following in the discussion:

"However, considering the limited sampling areas and time period, multiple-season measurements over spatially-distributed regions within the Arctic would be necessary for extending the data-driven study in its spatial and temporal representativeness".

For examining the impacts of intra-seasonal variability of Sentinel-2 data on our study, we replaced the original Sentinel-2 summer composite (June to August) by the July composite. For our study, the impact of Sentinel-2 intra-seasonal variability is relatively small, and the ALT results are highly consistent in their spatial patterns (R 0.91; RMSE 2.0 cm; Bias -0.57 cm; figure R1 below).

Fig R1. Comparisons between model predictions using Sentinel-2 summer composite (June to August; a) and July composite (b).

Further comments:

The capitalization in the section titles is not consistent.

**Response 2.8:** Thanks for the careful check. All section titles were formatted in the revision to ensure consistency in capitalization style.

**Comment 2.9**

L22: North Slope not Northern Slope

**Response 2.9: Corrected as below.**

"...within the North Slope of Alaska"

**Comment 2.10**

L62-63: You mention ALT derivation from LiDAR and from InSAR deformation signals driven by soil freeze—thaw. This connection needs further explanation as ground deformations do not directly translate to active layer thickness estimates without additional assumptions.

**Response 2.10:** As suggested, we added the following to further explain the connections between ALT estimation and measurements from LiDAR and InSAR.

"...more indirect measures of surface topographic features from Light Detection and Ranging (LiDAR) (Gangodagamage et al., 2014) and soil freeze-thaw (FT) driven land surface deformations from Interferometric synthetic aperture radar (InSAR) measures (Schaefer et al., 2015). The distribution of ALT over ice-wedge polygon landscapes was empirically inferred

using relevant topographic features quantified from LiDAR surface elevation measurements and data fusion approaches (Gangodagamage et al., 2014). In addition, the active layer heaves when frozen and subsides when thawed due to denser liquid water than ice. The seasonal vertical ground movement measured using InSAR is thus related to the ALT and ice/water content (Schaefer et al., 2015)".

**Comment 2.11**

L64ff: You state that low-frequency microwave measurements show strong potential for mapping ALT. However, the connection to active layer thickness is not entirely clear, since ALT cannot be directly measured with these observations. Could the authors elaborate on the underlying mechanism or clarify how these measurements can be translated into ALT estimates?

**Response 2.11:** As suggested, we elaborated the underlying mechanisms for using microwave to estimate ALT as below.

"Low-frequency (e.g., L- and P-band) microwave measurements are capable of penetrating through vegetation and soil layers and show strong potential for mapping active layer properties, including soil moisture and FT dynamics, organic matter content, and ALT (Tabatabaeenejad et al., 2014; Bakian-Dogaheh et al., 2025). This is due to the strong microwave sensitivity to the changes of soil dielectric properties, which are affected by soil moisture, texture, and freeze/thaw state (Kneisel et al., 2008; Du et al., 2019). In particular, the contrast in dielectric permittivity at the interface between thawed and frozen soil layers may lead to measurable microwave backscattering or reflection signals for estimating ALT (Kneisel et al., 2008)".

**Comment 2.12**

L75: The manuscript mentions "direct microwave sensing of soil profiles" for ALT. However, this is somewhat misleading, as ALT is not directly measurable from microwave observations.

**Response 2.12:** To be more rigorous, we deleted the words "Besides direct microwave sensing of soil profiles".

**Comment 2.13**

L92: Could you please provide information on the method on which the regional records are based?

**Response 2.13:** The sentence was expanded for providing more detailed description of the regional records as below.

"...and regional ALT records at 1-km resolution, which were generated using machine learning by combining in situ ALT observations with a suite of observational biophysical variables (Liu et al., 2024)".

**Comment 2.14**

L94: The numbering of the study plots starts at Plot 3 rather than Plot 1. Could you clarify why Plots 1 and 2 are not included or why the numbering begins at 3?

**Response 2.14:** Besides ALT measurements, our field experiment was also designed to conduct soil coring, soil moisture sampling, and surface roughness measurements over six plots (labeled 1 to 6). Intensive ALT sampling analyzed in this study was only performed on Plots 3, 4, 5, and 6. Here we used the same naming convention as our field experiment and the associated data archive.

**Comment 2.15**

L100: Is there a specific reason why the Canada Albers Equal Area Conic projection was chosen for Alaska? This projection results in scenes that are not north-oriented, with north arrows appearing consistently tilted. Could the authors clarify the rationale behind this choice?

**Response 2.15:** This study and field work are part of the larger NASA ABoVE campaign, which adopts the Canada Albers Equal Area projection for use and archiving of geospatial data products (<a href="https://above.nasa.gov/implementation\_plan/standard\_projection.html">https://above.nasa.gov/implementation\_plan/standard\_projection.html</a>). Accordingly, we used the ABoVE standard projection for defining the sampling plots and performing the following analysis.

**Comment 2.16**

Figure 1: It is unclear whether the black rectangles represent the actual footprints of the RGB images, as the displayed RGB images appear to be rotated relative to these rectangles.

**Response 2.16:** Thanks for the comment. In the revised Figure 1, the drone RGB images were clipped for showing the plot areas (black rectangles) only.

Figure 1: The study region encompasses an Arctic tundra area (68.6167°, -149.3167°; red dot in the inset) in the northern foothills of the Brooks Range, Alaska. The region consists of four intensively sampled plots (Plot 3, Plot 4, Plot 5, and Plot 6; 90 m x 90 m each) with their corresponding true-color RGB (red-green-blue) drone images displayed alongside for visual inspection only, and surrounded by a larger 5 km by 5 km study region (red rectangle) used for analyzing ALT scaling effects. The map was plotted using Canada Albers Equal Area Conic projection.

L146: The method chosen to aggregate the data is not specified and should be clarified.

**Response 2.17:** To be more clear, the sentence was revised as "The drone-based optical images and products were aggregated to both 0.1 m and 5 m resolutions through pixel averaging..."

L163: re-constructing or reconstructing

**Response 2.18:** We corrected the typo and used "...trained to reconstruct ALT patterns" in the revision.

**Comment 2.19**

L163-165: I find this sentence and Figure 3 somewhat misleading and suggest revisions to avoid confusion. In Figure 3, the arrow from the '0.1 m predictors' to the model could incorrectly suggest that the model was trained at 0.1 m, whereas it was actually applied at that resolution. This should be represented in a way that clearly distinguishes training from application/prediction to avoid any confusion. Additionally, the arrows from 'Aggregated to 10 m ALT' and 'Field ALT sampling' connect to the predictors, which does not seem correct and should rather lead into the models. Additionally, 'Scaling analysis' should be capitalized for consistency.

**Response 2.19:**

Thanks for the comment! The text and flowchart (Figure 3) were revised accordingly to avoid confusion. We also capitalized "Scaling analysis" for ensuring consistency. The revisions are also presented below:

"A data-driven Random Forest (RF) model was first trained to reconstruct ALT patterns at a 5-m resolution over the intensively sampled plots. This model was also used with 0.1-m predictors to generate the ALT maps at 0.1 m resolution. The 5-m ALT results were subsequently aggregated to 10-m resolution to train a second RF model, which was used to produce the 10-m ALT map over the surrounding region (Figure 3)".

Figure 3: An overall flowchart of the machine learning (ML) based active layer thickness (ALT) mapping and analysis.

Table 1: aspect (on the right side) should be capitalized for consistency

Response 2.20: Thanks for pointing out the inconsistency. "Aspect" was used in the revision.

**Comment 2.21**

Figure 4: The coordinate labels are difficult to read and should be enlarged for better readability.

**Response 2.21:** Figure 4 was re-plotted to ensure ALT measurements and predictions were only mapped over their overlapping areas for easier comparisons. The labels were also enlarged for better readability as suggested.

Figure 4. Comparisons of ALT spatial patterns derived from field measurements (a, d, g, j), 5-m machine learning outputs (b, e, h, k), and 0.1-m machine learning estimates (c, f, i, l) for the intensively sampled plots (grey shading indicates areas without sampling). The map was plotted using Canada Albers Equal Area Conic projection.

L247-249: This statement may be misleading, since NDVI (19.85%) is more important than aspect (17.85%) or slope (17.16%).

**Response 2.22:** The sentence was revised to be more rigorous as below.

"Terrain factors collectively form the most important control on the ALT distributions at 10-m resolution (elevation 29.88%, aspect 17.85%, slope 17.16%), while vegetation (NDVI 19.85%) and surface wetness (15.26%) conditions are relatively less important".

**Comment 2.23**

Figure 5: It would be helpful to also include elevation, since it is reported as the most important factor.

**Response 2.23:** As suggested elevation map was added to Figure 5 as shown below.

Figure 5: Comparisons between ALT distributions at 10-m resolution derived from machine learning (a), the synthetic ALT map at 0.1-m resolution (b), terrain aspect (c), slope (d), and elevation (e) over a larger 5 km x 5 km area surrounding the sampling plots (black/white rectangles). The map was plotted using Canada Albers Equal Area Conic projection.

L296-298: Only the change in R is reported for radar-based observations; it would be helpful to also show their importance relative to other predictors.

**Response:**

As suggested, feature importance information in radar-based ALT predictions was added as below.

"It is noted that additional radar (L-band 1.26 GHz) observations from UAVSAR helped to enhance the performance of the first RF model (e.g., R increased from 0.78 to 0.81), but were not used in the subsequent scaling analysis (section 3.1.2). For the features selected for radar-based ALT predictions, red-edge reflectance (16.67%), HV-polarized radar backscatter (16.04%), aspect (15.33%), and HH-polarized radar backscatter (15.04%) contributed most to the predictions, while the red band (13.34%), slope (11.99%), and green band (11.57%) observations were relatively less important. The sensitivity of radar backscatter to vegetation biomass, surface water bodies, and soil wetness likely enhanced the ALT estimation".